# Beyond surface tension-dominated water surface jumping

Xin Wang [1,6], Neng Xia [1,6], Chengfeng Pan [2,6] ✉, Jinsheng Zhao [1], Bo Hao[1], Lin Su[1], Dongdong Jin [3], Qingsong Xu [4], Xurui Liu [1], Xingyu Hou[1] & Li Zhang [1,5] ✉

Water surface jumping motions of semi-aquatic insects are primarily rely on surface tension-dominated jumping mechanism to achieve impressive jumping performance. However, this mechanism faces an inherent physical constraint: the propulsion force must remain below the threshold required to break the water surface, limiting efficient momentum acquisition. Herein, we present a water surface jumping strategy that addresses the limitations of surface tension-dominated mechanism. Our approach allows the engineered jumper to achieve a record-breaking jumping height of 18 body lengths (63 cm) and take-off velocity of 100.6 body length/s (3.52 m/s). This strategy is built on three key design principles: (I) superhydrophobic body for floating on water surface, (II) light-weight, high-power actuation module capable of providing significant propulsion force within an ultrashort time, (III) well-engineered momentum transmission system for efficient kinetic energy transfer. The developed soft jumper based on these design principles advances the development of water environment related robotics.

Insects have inspired researchers to develop miniature robots with various motion capabilities[1–8]. These robots, in turn, provide the biomechanical models that enhance the understanding of the locomotion behaviors of different species[9–16]. Semi-aquatic arthropods (e.g., water strider and springtails) have evolved unique mechanism for locomotion on the water surface, enabling them to jump both vertically and directionally with remarkable heights and speeds to evade predators or catch prey[1,2,17–19]. Their specialized locomotion relies on a surface tension-dominated jumping mechanism, which involves several key adaptations (Fig. 1a and b): (I) Surface tension utilization: semi-aquatic arthropods possess superhydrophobic bodies (e.g., the hairy, super-hydrophobic legs of water strider) and low body mass, allowing them to remain on the water surface without sinking by exploiting the surface tension of water[1,10,20]. (II) Anatomical feature and function: they

generate sufficient propulsion force to overcome the surface tension and detach from the water surface, facilitated by the specialized structures (e.g., the long, slender legs of water striders and the furcula of springtails) and rapid muscle contractions[2,21]. (III) Energetics and kinematics: high momentum transfer efficiency is achieved through ultrafast muscle activations (e.g., approximately <1 ms for springtails), sophisticated momentum transmission system and careful adjustment of propulsion force to remain just below the threshold needed to break the water surface. This allows semi-aquatic arthropods to use the water surface as a supportive medium for their movements without sinking[22–24].

Significant progress has been made in recent years in designing insect-scale water surface jumping robots by drawing insights based on these biological behaviors of surface tension-dominated jumping

[1]Department of Mechanical and Automation Engineering, The Chinese University of Hong Kong, Hong Kong, SAR 999077, P.R. China. [2]The State Key Laboratory of Fluid Power and Mechatronic Systems, College of Mechanical Engineering, Zhejiang University, Hangzhou, Zhejiang 310027, P.R. China. [3]School of Materials Science and Engineering, Harbin Institute of Technology (Shenzhen), ShenZhen, Guangdong, P.R. China. [4]Department of Electro-mechanical Engineering, Faculty of Science and Technology, University of Macau, Macau, P.R. China. [5]CUHK T Stone Robotics Institute, The Chinese University of Hong Kong, Hong Kong, SAR 999077, P.R. China. [6]These authors contributed equally: Xin Wang, Neng Xia, Chengfeng Pan. ✉e-mail: cfpan@zju.edu.cn; lizhang@cuhk.edu.hk

mechanism. Examples include latch-spring and motor-spring actuation systems, which possess hydrophobic body and can output adequate propulsion force enabled by the utilization of superhydrophobic materials and anatomy-based biomimetically structural design[1,25–32]. However, there remains a significant gap (on the order(s) of magnitude lower) between these engineered robots and their natural counterparts in terms of jumping height and take-off velocity (Supplementary Table 1). This gap stems from the inherent physical constraints of actuation strategies that rely on the water surface as a supportive medium. The driving force must remain below the threshold set by the surface tension of water. Exceeding this threshold diminishes the propulsion force (i.e., reaction force) and reduces momentum transfer efficiency due to the energy dissipation from breaking the water surface and generating splash[11,33]. This limitation causes that the ratio of propulsion force to body mass in these engineered robots is significantly smaller than in their natural counterparts due to their relatively heavy body and less sophisticated transmission structures compared to the highly efficient designs found in nature[11]. For example, Wang et al. developed a water surface jumper using lightweight soft materials actuated by an external magnetic field[34]. However, despite achieving a higher propulsion force-to-body mass ratio, this robot, along with other insect-scale water surface jumpers, continues to exhibit a notable disparity in jumping performance (Supplementary Table 1). The key issue lies in the inefficient energetics and kinematics, i.e., low momentum transfer efficiency. This inefficiency arises from the long actuation duration (typically is ~$10^1$ milliseconds) caused by the relatively unsophisticated design of actuation and transmission mechanisms compared to some of the nature species (e.g., springtails) with high-performance motion behavior, which rely on the fast muscle activations ( ~$10^0$ millisecond). Therefore, there remains necessary to develop a new actuation and transmission mechanisms to push the performance limits of engineered insect-scale water surface jumpers in order to match or even surpass the extraordinary jumping height and take-off velocity observed in natural species.

In this work, inspired by biological models and further developments, we introduce a water surface jumping mechanism for insect-scale robots that breaks through the physical constraints of surface tension-dominated jumping. This mechanism achieves jumping performance comparable to semi-aquatic arthropods (Supplementary Table 1) by incorporating three key design principles similar to the natural mechanism: (I) Superhydrophobic body: this ensures stable floating of the insect-scale jumper on the water surface, minimizing the energy loss during take-off by preventing excessive hydrodynamic interactions (Fig. 1c and Fig. 1e-top). In contrast, immersion would lead to significant liquid splash and energy dissipation (Fig. 1e-bottom). (II) Light-weight, powerful actuation module: capable of providing adequate propulsion force in an ultrashort time (comparable to rapid muscle activations in semi-aquatic arthropods), this allows the jumper to acquire momentum within a short time window (e.g., 0.6 ms in Fig. 1d-II). During this brief period, the water beneath the launching pad experiences limited deformation, allowing for a bounce force greater than the maximum reaction force that surface tension can provide, due to the incompressibility of water and hydro-dynamic interaction (Fig. 1e). (III) Well-designed momentum transmission system: a split-type design using soft tendon connections ensures efficient momentum transfer. This approach facilitates the momentum matching between the high-velocity actuation module (enabled by the substantial bounce force) and the low-velocity launching pad (due to the water resistance), maximizing energy transfer efficiency (Fig. 1c and Fig. 1f-top). In contrast, rigid connections would result in momentum mismatch and hinder the initial velocity acquisition (Fig. 1f-bottom).

With these design principles, we break through the limits of conventional water surface jumping, creating a leap toward new possibilities in biomimetic robotics by breaking the water surface,

redefining the jump (Table 1). We demonstrate this mechanism by developing a miniature water surface jumper (Fig. 1c and Supplementary Fig. 16), composed of three key components, including an ultrafast response actuation module as the driving source, a micro-pillar array-modified superhydrophobic launching pad to prevent sinking and efficiently transfer kinetic energy, and a pair of soft tendons serving as the momentum transmission system. The detailed actuation mechanism of the hydrogel actuator can be found in Supplementary Note 2. The proposed mechanism characterized with a high driving force-to-body mass ratio for greater acceleration and take-off velocity (Table 1), thus enables high-performance water surface jumping (Fig. 1). We compared the performance of demonstrated soft robot with other robot systems and insects in nature that are capable of water surface jumping motion (Fig. 2a), and the parameterized comparison in terms of jumping height and take-off velocity that consider the scaling (i.e., body length) factor as shown in Fig. 2b, the results shows that the demonstrated soft robot achieves the fastest take-off velocity (100.6 body lengths per second (BL/s), 3.52 m/s) and the highest launch height (18 BL, 63 cm) of any engineered water surface jumper reported to date, rivaling the performance of pygmy mole crickets, known for their exceptional jumping capabilities among semi-aquatic arthropods (Supplementary Table 1).

## Results

### Design principles for high-performance water surface jumping

In surface tension-dominated jumping, the reaction force from the water surface (i.e., driving force) must remain below the surface tension threshold to maximize the kinetic energy transfer to the jumper instead of water by avoiding the water surface broken, which can result in significant splashing. Once the water surface is broken, the drag force from the disrupted water leads to substantial kinetic energy loss for the jumper. In contrast, the mechanism we propose does not require the water surface to remain unbroken during the jumping process. The soft connection structural design makes the acquisition of actuation force mainly rely on the interaction of solids (i.e., rebound force when the actuator hitting the launching pad) rather than the reaction force of the surface tension. Thus allowing the significant increase of the actuator's power output for initial momentum acquisition. These design principles allow greater flexibility in energy transfer, enhancing overall performance without the constraints imposed by traditional designs.

The most critical design criterion for the proposed water surface jumping mechanism is an efficient momentum transmission system that maximizes the initial velocity while minimizing the kinetic energy loss due to the broken of water surface. In this work, we use a split structure design with a pair of soft tendons as the momentum transmission system. As shown in Fig. 3a, when the actuation process begins, the actuation module and the superhydrophobic pad experience equal and opposite forces, resulting in the upward movement of the actuation module and the downward movement of the superhydrophobic pad. Because of the soft tendon connection (i.e., no drag force), the initial upward motion of the actuation module encounters only air resistance and gravity, allowing it to achieve a high initial velocity with minimal reduction (Fig. 3b). In contrast, the superhydrophobic pad moves downward into the water, facing multiple resistances, including hydrostatic force, bounce force and surface tension. These resistances lead to a more rapid velocity reduction for the superhydrophobic pad in water. As the tendons straighten due to the velocity mismatch between the actuation module and the superhydrophobic pad (Fig. 1d-III), the total body velocity ($v_{total}$) can be determined using the momentum theorem[35]:

$$v_{total} = \frac{m_b v_b + m_p v_p}{m_b + m_p} \qquad (1)$$

where $m_b$ (266 mg), $v_b$, $m_p$ (144 mg), and $v_p$ are the masses and velocities of the robot's body and superhydrophobic pad, respectively. The larger driving force from the actuator results in higher upward velocity of the body ($v_b$) (i.e., actuation module) and a more rapid decay of the superhydrophobic pad's downward velocity ($v_p$) until it approaches zero. The difference in the velocity decay rate in opposite directions between $v_b$ and $v_p$ results in an upward movement and take-off velocity of whole jumper (as described by Eq. (1)). In other word, the momentum transfer strategy enabled by the soft connection allows for a driving force that exceeds the threshold required to break the water surface, enabling the actuator to maximize its output and achieve a higher initial velocity without concern for surface disruption (Fig. 3b). Although the jumping motion without the removing process of launching platform from water can achieve more attractive motion performance (Supplementary Fig. 17), in this work, we still take the launch pad as a part of a water surface jumping robot which required all components leaving from water to accomplish a water surface jumping motion. We also calculated the take-off velocity after the soft tendons straighten (after 6 ms, as shown in Fig. 1d-III) from the

recorded maximum jumping height (Supplementary Fig. 8a and Supplementary Movie 2) by modeling the post-take-off motion of the jumper as deceleration due to the gravity and the air drag force ($F_{drag} = C_d \rho \pi r_0^2 v_f^2/2$, where $v_f$ is the flying velocity)[36,37],

$$v_{take-off} = \sqrt{\frac{\left[e^{\frac{hC_d\rho\pi r_0^2}{m}} - 1\right] \times 2mg}{C_d\rho\pi r_0^2}} \quad (2)$$

Here, $C_d$ is the drag coefficient, $h$ is the maximum jumping height, $\rho$ is the density of air, $r_O$ is the basal radius of the soft robot, $m$ is the total mass of the jumper. For the demonstrated jumper depicted in Supplementary Fig. 18a, the calculated take-off velocity for the water surface jumping motion is 3.52 m/s (100.6 BL/s). The released energy by the hydrogel actuator applied to the floating pad and the jumping energy utilization efficiency are estimated to be 5.16 mJ and 49.22 %, respectively (see calculation details in Supplementary Note 3). It is

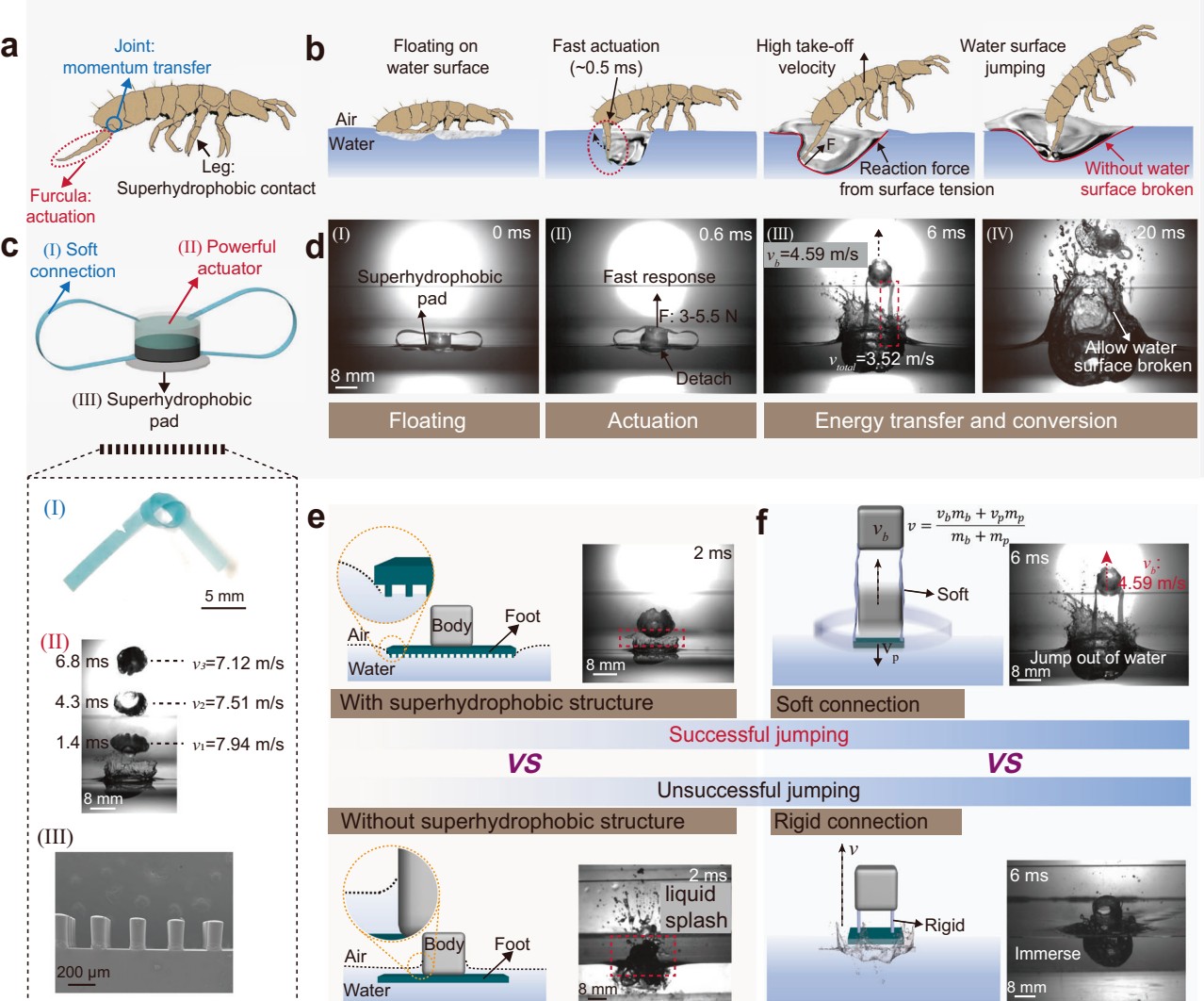

**Fig. 1 | Key design principles of developed water surface jumping mechanism.**
**a** Schematic illustration of the functional parts of the springtails and (**b**) its water surface jumping motion based on the surface tension-dominated mechanism.
**c** Schematic illustration of the functional modules of the water surface jumping soft robot based on proposed design principles and (**d**) the high-speed camera images obtained from Supplementary Movie 1 showing the take-off process of the soft robot. Note that the $v_b$ and $v_{total}$ in the image is the speed of the actuation module and the take-off velocity of the whole soft robot, respectively. **e** The comparison of the robot with and without the superhydrophobic contact with water. **f** The comparison of the robot with soft connection and rigid connection.

**Table 1 | The comparison of the previously reported surface tension-dominated mechanism employed by animals and artificial jumpers and our proposed water surface jumping mechanism**

|  |  | Superhydrophobic contact with water | Actuation time | $F_{max}/m_{total}$ (N/kg) | Allow water surface broken |
|---|---|---|---|---|---|
| Surface tension-dominated mechanism | Animals (Springtails[2]) | √ | $< 10^0$ ms (0.5 ms) | $2.92 \times 10^3$ | × |
|  | Artificial (Water strider bionic robot[1]) | √ | $10^0 - 10^1$ ms (15 ms) | $1.36 \times 10^2$ | × |
| This work |  | √ | $< 10^0$ ms | $1.34 \times 10^4$ | √ |

Note the detailed calculation of $F_{max}/m_{total}$ can be found in Supplementary Note 1.

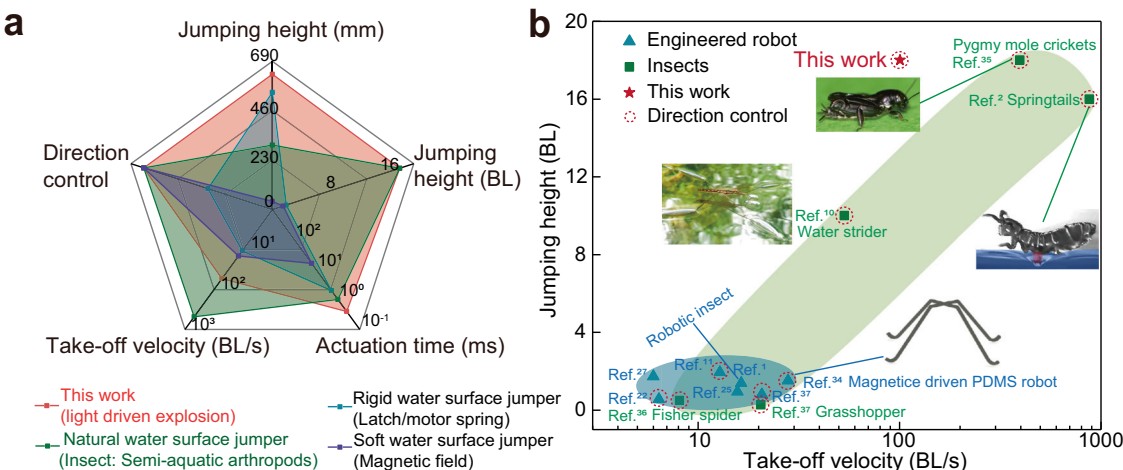

**Fig. 2 | Performance of the demonstrated water surface jumper based on proposed mechanism. a** Radar chart illustrates the performance comparison for the water surface jumper based on the proposed mechanism with natural and engineered water surface jumping system. **b** Jumping height and take-off velocity comparison of the soft robot in the current study with previously reported engineered water surface jumping robots and insects[1,2,10,11,22,25,27,34,43–45]. Note that the take-off speed shown in Fig. 2 are obtained from Supplementary Movie 2. Figures reproduced with permission from: (left inset), ref. 10, under a Creative Commons Attribution License 4.0 (CC BY 4.0); (up inset), ref. 43, Elsevier; (bottom inset), ref. 34, under a Creative Commons CC BY license. (right inset), ref. 2, under a Creative Commons Attribution-NonCommercial-NoDerivatives License 4.0 (CC BY-NC-ND).

important to note that the energy loss primarily occurs during the energy conduction process and in the downward movement of superhydrophobic pad as it overcomes hydrodynamic forces from the water. The similar performance of our split-type soft robot jumping on solid glass surface (78 cm, 22.3 BL) further demonstrates the high momentum transfer efficiency due to the limited energy dissipation (Supplementary Fig. 18b and Supplementary Movie 2). In contrast, a jumper constructed with rigid connections exhibited limited take-off speed and unsuccessful water surface jumping due to the inefficient energy transfer system (Fig. 3b and Supplementary Fig. 19).

Based on the previous analysis, the proposed water surface jumping mechanism overcomes the physical constraints of surface tension-dominated mechanism, allowing the driving force to exceed the threshold determined by the surface tension of water. This enables the use of a powerful actuation module with fast-response characteristic and large driving force to generate high acceleration and initial velocity, thereby improving the jumping performance. In this work, we adopt our previously reported hydrogel actuator[38] in the fabrication of water surface jumper as a demonstration, which relies on the strain energy accumulation during deformation and instant release through fracture to realize the powerful actuation. We measured the energy release time and compared with the conventional representative spring actuators (Supplementary Fig. 20), which is crucial for achieving higher energy transfer efficiency from potential energy to kinetic energy. Following the principle of power amplification, under the same energy accumulation conditions, a shorter energy release time leads to a greater power density output, allowing the jumper to achieve larger impulse and initial kinetic energy[38]. The energy release time of the hydrogel actuator and

spring actuators are measured using a high-speed camera, with the results shown in Fig. 3c, d. The hydrogel actuator exhibits a much faster energy release time of 0.6 ms, compared to 2.1 ms for hard springs and 1 ms for soft springs due to the fracture driven power amplification mechanism as we presented in previous work[38], details of the further optimization of the hydrogel actuator can be found in Supplementary Note 4. In addition to the energy release time, the total body mass of the jumper is strictly limited by the bearing capacity of water surface, necessitating an actuator with a high output driving force-to-body mass ratio for greater acceleration. We measured output force ratio of this hydrogel actuator and compared it to spring actuators, as shown in Fig. 3d. The results indicate that the hydrogel actuator significantly outperforms spring actuators in terms of the driving force-to-mass ratio. Notably, the force $F$ of the hydrogel actuator was detected on a force sensor, while for the springs, the maximum force was detected during the extreme compression, as illustrated in the inset of Fig. 3d and Supplementary Fig. 21. Consequently, the hydrogel actuator was chosen to demonstrate the proposed water surface jumping mechanism due to its combination of ultrafast energy release time and high driving force output. The current hydrogel actuator has the shortcoming of single time use despite this can be compensated by simple large-scale fabrication (Supplementary Fig. 22). Furthermore, it is important to note that this actuator serves as a convenient and typical candidate, other rigid or elastic actuation structures with faster response times and greater force-to-mass ratios can also be utilized in jumpers based on the proposed water surface jumping mechanism.

The third design principle involves the floating method of the water surface jumper. The initial stable floating state is indispensable

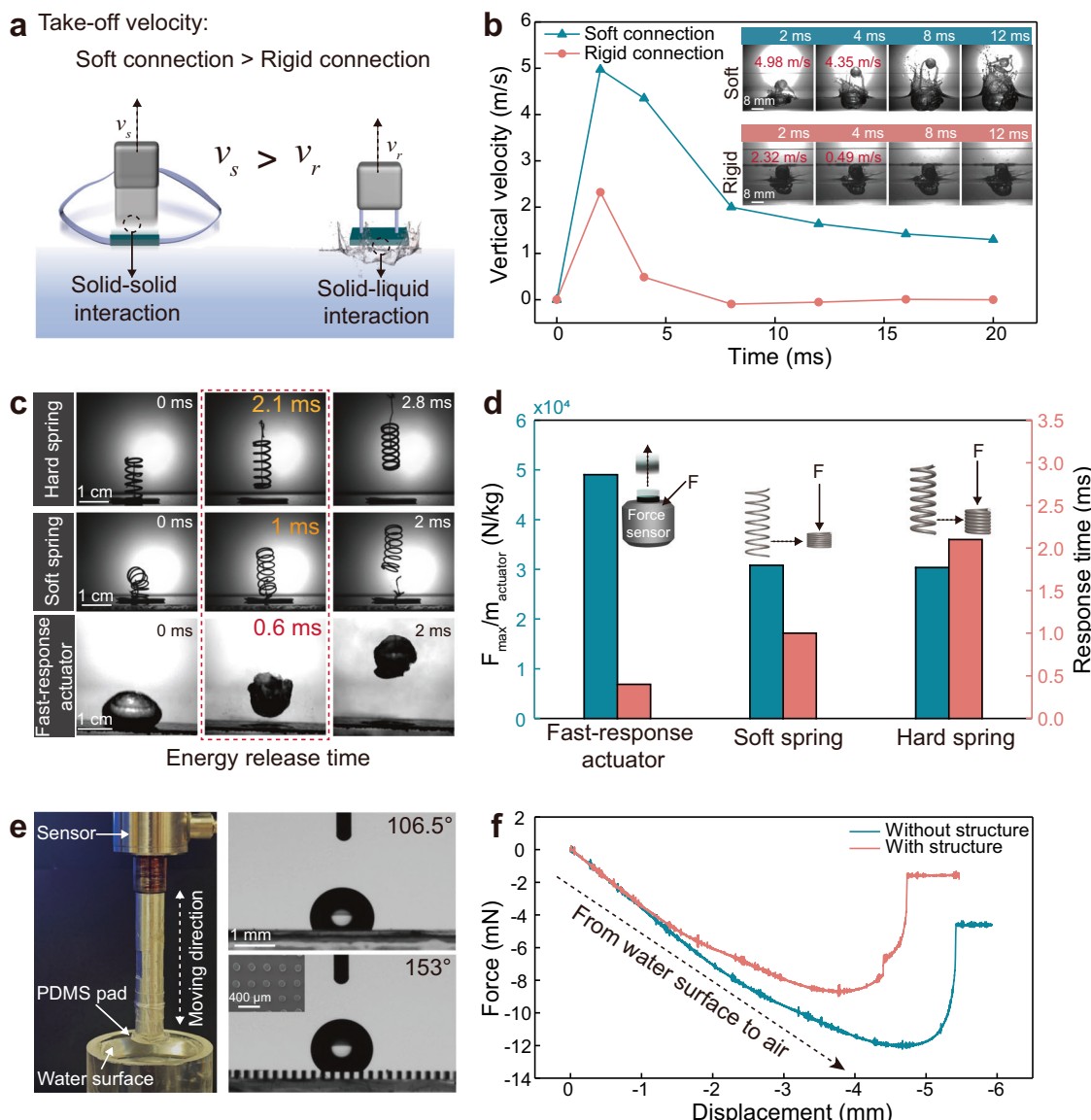

**Fig. 3 | Comparison and analysis of three design principles of the proposed mechanism. a** Schematic illustration the initial momentum acquisition of the soft and rigid energy transfer system. **b** The take-off velocity change of the soft and rigid connection robots. Note that the take-off velocity is the whole speed of the soft robot that obtained from Supplementary Movie 1. **c** The high speed camera images illustrate the response time of the representative actuators (i.e., hard spring, soft spring and fast-response actuator) launching from glass substrate. Note that the initial stresses are set to approximately 5.5 N. **d** The comparison of the output force/mass ratio and response time of the representative actuators launching from glass substrate. Note the detailed calculation of $F_{max}/m_{actuator}$ can be found in Supplementary Note 1. **e** The image illustration of the experimental setup, the PDMS pad with and without micropillar array mounted on a force sensor and slowly pulled out from the water surface at a constant speed of 0.1 mm/s (left). The contact angle of the PDMS pad with and without micropillar array (right). Inset: SEM image of the top view of the micropillar array. **f** The surface tension change during the process of the PDMS pad pulled out from water surface.

that relies on the surface tension of water. In this work, we used PDMS as a hydrophobic material to fabricate the floating pad that supports the actuation module while providing solid interaction for initial momentum acquisition. Additionally, we fabricated micropillar arrays (diameter = 90 μm, height = 200 μm) on the PDMS pad to enhance its superhydrophobicity (Fig. 3e and Supplementary Fig. 23). To quantify the effect of surface tension on the jumping motion, we mounted the PDMS pad on a force sensor and pulled it upward from the water surface at a constant speed of 0.1 mm/s. The results, shown in Fig. 3f, indicate that the detected force increases during the upward motion of the PDMS pad, which can be attributed to the change in the air-water interface angle as calculated by the equation $F = F_\sigma \sin\varphi$. Here $F_\sigma$ represents the surface tension, calculated as $F_\sigma = \gamma L$, where $\gamma$ is the surface tension coefficient, $L$ is the perimeter of the contact area, and $\varphi$

is the angle of the air-water interface due to the deformed water surface (Supplementary Fig. 24). Once the limit of surface tension is reached, the detected force drops sharply when the PDMS pad exits the water (Fig. 3f and Supplementary Fig. 25). Notably, the presence of micropillar arrays on the PDMS pad significantly reduces the surface tension at the moment of detachment (8.7 mN vs. 12.05 mN). These findings demonstrate that the superhydrophobic contact not only ensures stable floating on the water surface but also minimizes energy loss during detachment in the take-off process.

For the current design, the jumping performance of the soft robot can be influenced by many factors, e.g., light intensity, power output of the hydrogel actuator, hinge length and diameter of the launching pad, etc. We have tested these influencing factors and the detailed discussion can be found in Supplementary Note 5.

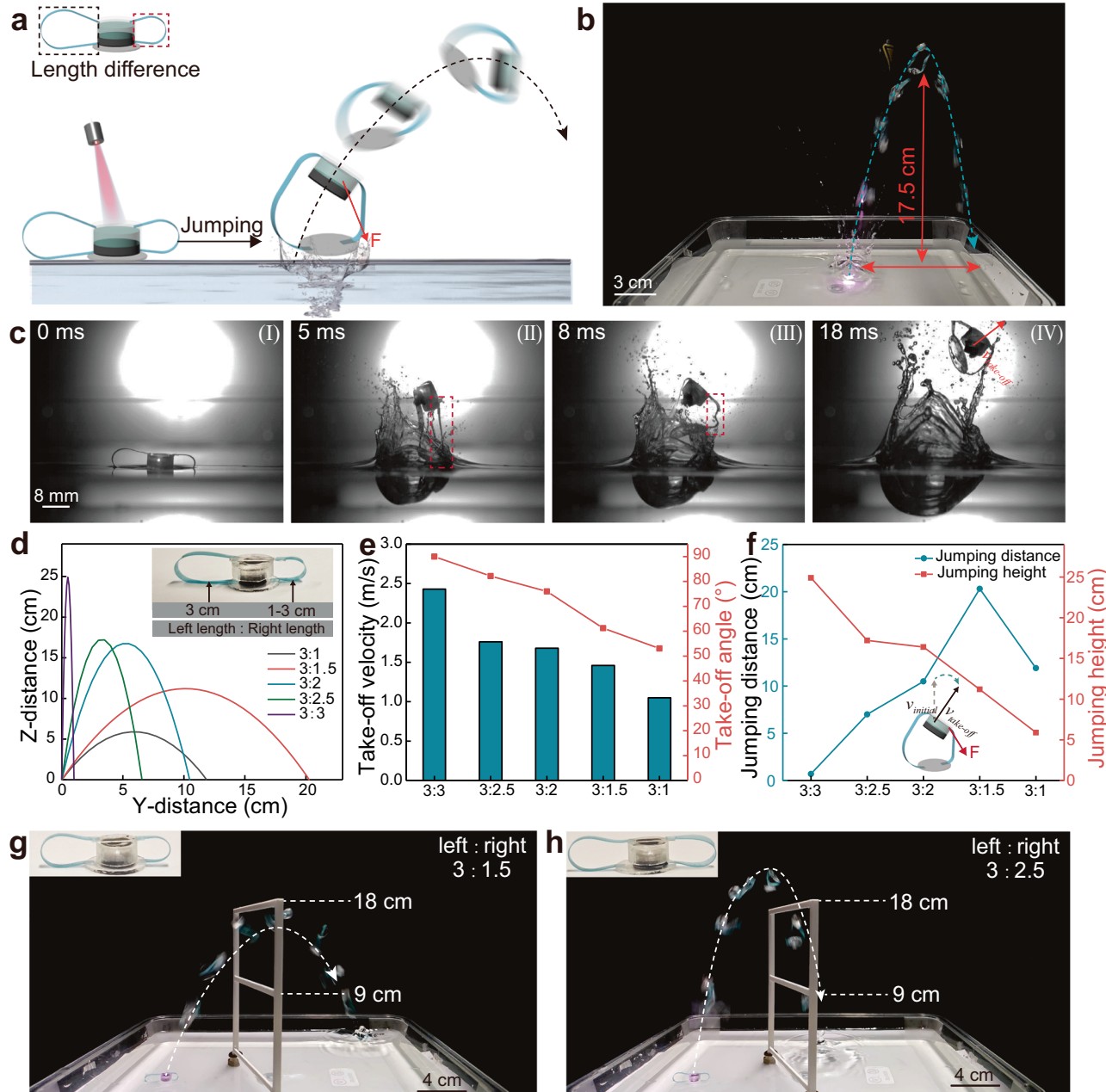

**Fig. 4 | Motion control of the water surface jumping soft robot. a** Schematic illustration showing the direction control principle of the water surface jumping soft robot. **b** Overlaid images obtained from Supplementary Movie 3 showing the directional water surface jumping motions. **c** High-speed camera images showing the directional take-off process. Note that the tendon length of the left: right is 3: 1.5 (cm). **d** The jumping trajectories of the soft robots with different left/right tendon length ratios. **e** The take-off velocity and take-off angle of the soft robots with different left/right tendon length ratios. **f** The jumping distance and jumping height of the soft robots with different left/right tendon length ratios. **g, h** The overlaid images indicating the soft robot with representative left/right tendon length ratio across railings with different heights. Note that all the data captured in Fig. 4 are with the light intensity of 2.8 W (4.4 W/cm²).

## Motion control of the water surface jumping soft robot

In addition to the vertical water surface jumping motion of the symmetrical structural soft robots, the directional jumping motion can be realized through asymmetrical structural design. As shown in Fig. 4a, the left and right tendon are designed with different lengths (i.e., the change of left/right tendon length ratio), this results in the soft robot can jump to different directions depending on the towards of the shorter tendon. As shown in Fig. 4b and Supplementary Movie 3, the soft robot jumps to the right that is the direction of the shorter tendon towards. The potential reason is that the asymmetrical structural design through influencing the take-off velocity to change the jumping direction (Fig. 4a). We use high-speed camera to record the directional

jumping motion of the soft robot as shown in Fig. 4c, the actuation module of the soft robot first accelerates until the shorter tendon (right tendon) be straightened due to the reaction force from the hydrogel actuator hits the PDMS pad, the longer tendon (left tendon) is still in unstraightened state at this time (Fig. 4c-(II)). The right side of the PDMS pad begins to experience upward pulling force, but the left side maintains the initial force condition. This, in turn, induces a downward force from the drag force and hydrodynamic force of the water that is applied on the right side of the soft robot's body (Fig. 4c-(II)), which results in the direction of the take-off velocity changed and thus realizing the directional water surface jumping motion behavior (Fig. 4c-(IV)). The vertical jumping motion on the glass surface of the

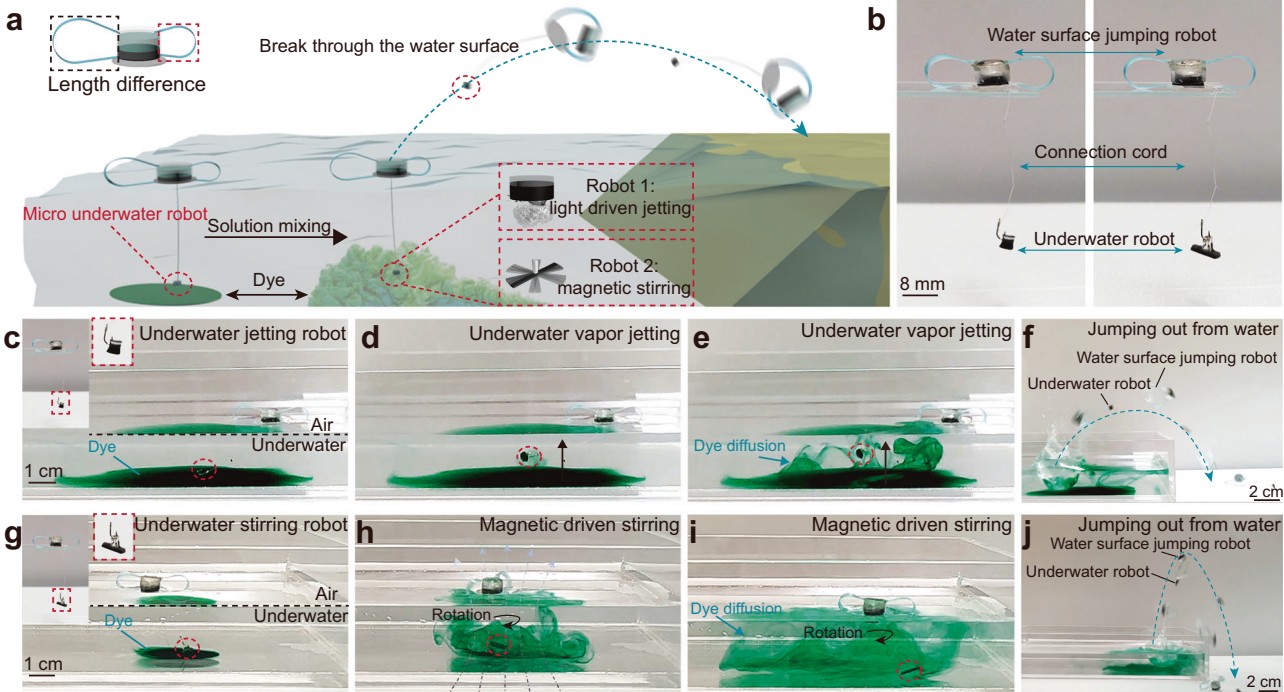

**Fig. 5 | Water surface soft jumper help underwater micro-robot to break through the air-water surface for recycling. a** Schematic illustration showing the water surface jumping soft robot help the underwater solution mixing robots break through the water surface after task performing. **b** The images showing the integration of the underwater microrobots and water surface jumping soft robot. **c** Photographic sequence obtained from Supplementary Movie 5 of the (**c**–**f**) underwater micro robot use light driven vapour jetting and (**g**–**j**) magnetic driven stirring to perform solution mixing task and followed recycle process.

soft robot with asymmetrical structural design indicating the force that deflects the jumping direction comes from the hydrodynamic forces (Supplementary Fig. 26).

The asymmetrical structural design induced take-off velocity changes also can be utilized to adjust the jumping trajectories. We fabricated a series of soft robot with different left/right tendon length ratio as shown in Fig. 4d, the parabola trajectory of the representative soft robot can be determined by recording the jumping process. The results show that the take-off velocity is highly depended on the left/right tendon length ratio, the magnitude of the initial velocity and the take-off angle are decreased with the increase of the left/right tendon length ratio (Fig. 4e), which finally realized the adjustment of the jumping trajectories (Fig. 4d and Supplementary Movie 4). Specifically, the closer of the left and right tendon lengths, the soft robot are more tends to vertical jump, and the greater the tendon length difference, the soft robot exhibits a longer jumping distance but lower jumping height (Fig. 4d and f). Note that the excess short of one side tendon length will induce the performance degradation, thus exhibiting the shorter jumping distance and jumping height (the soft robot with left/right tendon length ratio of 3: 1 shown in Fig. 4d and f). The jumping trajectory control strategy based on the change of left/right tendon length ratio is visually demonstrated by the directional water surface jumping behavior of two presentative soft robots with the ratio of 3: 1.5 and 3: 2.5 shown in Fig. 4g and h, respectively. The soft robots are able to jump through railings with different heights actuated by NIR light, note that the shorter tendon of the soft robot is oriented perpendicular to the plane of the railing in the initial state.

## Application for underwater micro-robot recycle

Micro robots exhibit various application potentials in complex environment, e.g., underwater environment. Due to the physical small scale and light weight, micro underwater robot shows advantages in navigating and task performing through remotely untethered control

ways[39]. However, the main challenge of recycling these microrobots is the resistance and huge surface tension at small scale during breaking through the air-water interface[40]. Based on the high-efficient energy transmission and loading capability of the current water surface jumper, we demonstrate a robotic system for recycling of the underwater microrobot. The representative example is the recycle of the underwater solution mixing microrobot as shown in Fig. 5a, among others, the solution mixing task can be realized by light driven vapour jetting of hydrogel microrobot and the magnetic driven stirring of PDMS/NdFeB microrobot, respectively. These underwater microrobots are connected with water surface jumper realize the function of anchoring with each other to avoid the deviate in target area (Fig. 5b). As a proof of concept, the underwater micro robots first perform the solution mixing task controlled by light or rotation magnetic field, respectively (Fig. 5c-e and Fig. 5g-i), and then the recycle of the underwater micro robots can be realized through activating the water surface jumper (Fig. 5f, j and Supplementary Movie 5). In addition, we also explored the application potentials in the field of the oily pollutant removal in water surface environment, as shown in Supplementary Fig. 27a-b and Supplementary Movie 5, a PDMS magnetic sponge module was integrated into the soft robot, enabling the soft robot to be navigated by the magnetic field and automatically absorb the floating oily pollutants (Supplementary Fig. 27c-e). After completing the oily pollutant collection, the soft robot jumps out of the water surface (Supplementary Fig. 27f). The comparison of the PDMS sponge shown in Supplementary Fig. 28 indicates the application potentials in environment detection and protection. These experiments provide an indirect remote-control strategy and validate the application potential of the current water surface jumpers for water environment related applications.

## Discussion

In summary, we have introduced a water surface jumping strategy beyond the surface tension dominated mechanism for insect-scale

robots that harnesses three key design principles to break through the physical constraints of current jumping mechanism (Supplementary Note 6), including a superhydrophobic body for floating on water surface, a light-weight powerful actuation module that can provide adequate propulsion force in an ultrashort time (muscle activation timescale) for large initial momentum acquisition, and a soft momentum transmission system for highly efficient kinetic energy transfer. This water surface jumping mechanism endows the demonstrated soft robots with exceptional jumping performance in terms of jumping height (18 body lengths (BL), ~63 cm) and take-off velocity (100.6 BL/s, 3.52 m/s), outperforming what has previously been demonstrated by engineered system and almost all the insects with water surface jumping motion in nature. The proposed water surface soft jumping robot can be further designed to achieve controllable directional jumping to over the obstacles and applied to help the recycle of underwater micro-robotics.

## Methods

### Materials
N-Vinyl-2-pyrrolidone (NVP), acrylic acid (AA), N,N'-methylenebisacrylamide (MBA), 2,4,6-trimethylbenzoyl-diphenylphosphine oxide (TPO), polyvinylpyrrolidone (PVP), sodium dodecyl sulfate (SDS), ferric oxide ($Fe_2O_3$) and isopropyl alcohol (IPA) were purchased from Aladdin Chemicals. The oil soluble dye, black ink, hard magnetic particles neodymium-iron-boron (NdFeB), Polydimethylsiloxane (PDMS), Ecoflex 00-30 and graphene (Gr) were purchased from Alibaba Co. Ltd, China.

### Preparation of hydrogel actuator
In a typical experiment, 60 mg water-soluble TPO nanoparticles and 25 mg MBA were first dissolved in 9 mL deionized water, the water-soluble TPO nanoparticles were prepared by the method reported in the literature[41]. Next, 3 mL AA, 3 mL NVP and 25 mg graphene were added under stirring and then ultrasonic treatment for 10 min to obtain a homogenous dispersion. The hydrogel precursor contained graphene (solution A) were successfully prepared after placed in vacuum 5 mins to remove oxygen. Note that the graphene can be replaced by other materials with photo-thermal ability (e.g., $Fe_2O_3$ nano particles and black ink as shown in Supplementary Fig. 29). The transparent hydrogel precursor (solution B) was prepared use same procedure without the introduction of graphene. The hydrogel film was prepared by layer-by-layer method. Specifically, the hydrogel precursor contained graphene (solution A) were polymerized as bottom layer in a 3D printed mold under irradiation of UV light for 5 mins. Next, the solution B was introduced and polymerized as upper layer through irradiated with UV light for 4 mins, after that, the hydrogel films that the bottom layer contained graphene and upper transparent hydrogel layer were combined, and the hydrogel actuator were successfully fabricated through cutting by circle shape punch with the diameter of 8 mm. In the current study, unless otherwise mentioned, the water content, graphene content and light intensity were 60 wt%, 0.17 wt% and 6.3 W (9.91 W/cm$^2$), respectively.

### The fabrication of light driven water surface jumper
The hydrogel actuator-load module was fabricated by template method with 3D printed molds, the parameter of the mold was shown in Supplementary Fig. 30. In a typical experiment, the PDMS with 10:1 weight ratio of the monomers and curing agents were stirred for 3 mins, then poured to the mold and vacuum degassing for 10 mins, the mold filled with precursor was then placed in oven (60 °C) for 2 h until platinum catalyzed polymerization process finished, the transparent PDMS actuator-loaded module successfully prepared after demolding and sealing process.

For the fabrication of the PDMS pad with superhydrophobic structure, the PDMS with 10:1 weight ratio of the monomers and curing

agents were mixed and vacuum degassing, and poured on the silicon mold with microporous structure, the PDMS film with micropillar array structures were successfully fabricated after curing and demoulding process, the PDMS pad can be obtained through cutting by punch with different diameters.

The PDMS tendons were fabricated through spin coating method. First, the mixed PDMS solution with introduction of dye were deposited on the glass at 600 rmp for 30 s, then, the glass was placed on the heating plate to fabricate the PDMS film. The tendon with different length can be successfully fabricated after cutting.

After all the modules were prepared, the PDMS mixture solution were used to bond them together, the water surface jumping soft robot were successfully fabricated after manual loading of the hydrogel actuator.

### The fabrication of magnetic PDMS sponge
The oil absorption module (PDMS sponge) was fabricated by emulsion method based on the literature[42]. Specifically, 12 g paraffin oil and 10 g NdFeB micro particles were added to the 10 g mixed PDMS solution (10:1) under vigorous stirring. Then, 70 g distilled water was slowly added dropwise under stirring until the PDMS-water emulsion was formed. Next, the PDMS-water emulsion was polymerized in the oven (70 °C for 2 h). After polymerization, the PDMS sponge was washed with ethanol three times and dried in the oven. The magnetic oil absorption module was successfully fabricated after cutting and integrated on the soft robot.

### Characterization
The high-speed images and video were recorded by SA-Z Photron high speed camera with the frame rate of $1\times10^4$ frame per second (fps). The light source was 808 nm NIR light from Beijing Leiyuan Technology Co., Ltd. Unless otherwise mentioned, the laser power and distance between the light source and soft robot were 6.3 W (9.91 W/cm$^2$) and 25 cm, respectively. The SEM test was performed on JSM-7800F scanning electron microscope. The FTIR spectra were performed on Thermo Nicolet Nexus 670. The tensile experiments and toughness measurement were performed on a MACH-1 system. The forces were recorded by a force sensor (Nano 17, ATI). The temperature change and infrared thermal images were obtained by using a FLIR-A300 camera.

## Data availability
All data are available in the Article or its Supplementary Information. Source data are provided with this paper.

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

## Acknowledgements

This work was supported by the Hong Kong Research Grants Council (RGC) with project nos. R4015-21, C1134-20GF, RFS2122-4S03 and STG1/E-401/23-N (L.Z.); We also thank the support from the SIAT-CUHK Joint Laboratory of Robotics and Intelligent Systems and the Multi-scale Medical Robotics Centre (MRC), InnoHK, at the Hong Kong Science Park.

## Author contributions

X.W., C.P. and L.Z. proposed concept and designed the research. X.W. performed the experiments and analyzed data. X.W., C.P., N.X. and Q.X. conduced the model analysis. J.Z., B.H., L.S. and D.J. given the suggestions in designing experiment and data analysis. X.L. and X.H. helped the data collection. L.Z. and C.P. supervised the research. All authors discussed the results and wrote the manuscript.

## Competing interests

The authors declare no competing interests.
