## [Transparent Peer Review file · Nature Communications]

Beyond surface tension-dominated water surface jumping

Corresponding Author: Professor Li Zhang

Version 0:

Reviewer comments:

Reviewer #1

(Remarks to the Author)

Reviewer #2

(Remarks to the Author)

This study introduces a novel water surface jumping mechanism for insect-scale robots, inspired by biological models. The design achieves jumping performance comparable to semi-aquatic arthropods by incorporating three main principles: a superhydrophobic body, a lightweight powerful actuation module, and an efficient momentum transmission system. Altogether, these design principles enabled the robot to overcome surface tension constraints, achieving effective water surface jumping. Overall, this study presents an innovative approach to insect-scale water surface jumping mechanisms, which is intriguing for the field and has potential applications in environmental monitoring or health monitoring of aquatic ecosystems through scalable data management and analysis in the cloud. Therefore, this manuscript is acceptable for this journal after addressing the following comments.

1. The manuscript employs a hydrogel actuator, referencing the fracture-driven power amplification mechanism from "Fracture-driven power amplification in a hydrogel launcher" (Nat. Mater. 23, 1428-1435, 2024). Could the authors elaborate on any specific design or structural modifications made to the hydrogel actuator in this study compared to the one described in the 2024 Nature Materials paper? Understanding these design differences or optimizations would provide valuable insights into how this actuator has been tailored to meet the unique demands of water surface jumping in microrobotics, particularly in terms of response time, energy release, and force-to-mass ratio.
2. To assess the hydrogel actuator's suitability for repeated use in microrobotics, it would be helpful if the authors could clarify how swelling impacts its performance, specifically in terms of response time, force output, and energy efficiency. Additionally, has the authors conducted tests on performance stability across multiple jumping cycles to ensure durability and consistent functionality? Providing this information would strengthen understanding of the actuator's reliability for continuous applications.
3. To enhance the robot's versatility and expand its usability for sustained microrobotic applications on or near water surfaces, would the authors consider integrating the jumping mechanism with additional locomotion modes, such as directional movement or swimming? While the manuscript demonstrates an impressive water surface jumping capability comparable to semi-aquatic arthropods, combining jumping with other movement types could support more complex or continuous operations in water-based environments.

Reviewer #3

(Remarks to the Author)

The paper presents an insect-sized water-jumping robot that can jump on a water surface with a high drag. The research topic is interesting, and the results show an impressive jumping performance of the robot. A novel hydrogel-type actuator and hydrophobic structure on the body seem to be the main components for jumping on water. As a potential application, the

authors show that the robot takes a micro underwater robot out of the water for recycling. The demonstration of the potential application is also interesting and meaningful. However, the significant contribution and novelty of the paper are unclear for the readers to be able to learn and understand the mechanism happening on the water surface.

In terms of the scientific analysis of this research, the paper should describe the design principles that can manage the drag from the water surface such as shape, force profile, and actuation speed, for achieving the desired performance for the robot. The maximum performance of the robot is recorded, but the dynamics of the mechanism are not fully established so that we can understand the design objective. As far as I understand from this paper, it seems that the high power of the actuator is the main design principle.

In terms of technical improvement, the actuation mechanism with hydrogel is used as the main actuator, which is introduced in the previous study. The hydrophobic structure on the body was also published in multiple papers. A breakthrough enabling technology for this study is unclear. I understand many engineering techniques should be applied to make this robot, but it is not enough to convince of the significance of the research to be published in Nat. Comm.

Before going to the detailed review of the manuscript, these major concerns should be convinced to consider the publication of the paper. The research topic and approach are good, but scientific findings and technical improvement are required for the broad readership of the journal.

Version 1:

Reviewer comments:

Reviewer #1

(Remarks to the Author)

The jumping mechanism is now quite clearly explained and supported by Supplementary Figs. 6 -7.

Please explicitly indicate the cases where F_{\max}/m and not F/m ratio was presented.

I like all the new figures. They do help to understand the volume of experimental work performed by the authors.

Comment 15: Second part of Eq. 8 is wrong (see attached equation). It seemed that you've used diameter instead of radius in the equation for the truncated cone volume. The actual volume is 4 times smaller.

Eqs. 10 and 12 give the upper bound of the potential energy of displaced water.

I have no further comments.

Reviewer #2

(Remarks to the Author)

Authors addressed and discussed the raised issues in the light of the reviewer's comments. In the revised manuscript, the authors emphasized explanation of the insect-scale water surface jumping mechanisms for soft robots. In addition, they have adequately supplemented with references and requested data to support their claims. The manuscript now demonstrates a more comprehensive view of optimizations in terms of structure design, swelling stability of the actuator, and demonstration of the robot providing detailed information. I recommend the paper could be published in Nature Communications without further change.

Reviewer #3

(Remarks to the Author)

Authors addressed issues raised by the reviewer.

Here are some comments that can improve the quality of the paper.

- In Fig.1, the illustration of the springtail jumping sequence is redundant information in this research. It is hard to find relevance between this mechanism and the springtail jumping
- The authors say that the soft connection is a key design factor for the robot. It should be proven with theoretical modeling for a clear understanding how the stiffness of the connection affects to the jumping performance.
- Is it possible to give any design optimization for the high jumping on the water surface based on the power of the actuator?

Reviewer #1:

The manuscript entitled “Beyond surface tension-dominated water surface jumping” presents an explosive steam/water blast driven jumping of a polymer block from a platform floating on the water surface and connected elastically to the block. The polymer lifter flies up with the platform from the water surface to the height of up to 18 body lengths after it has been heated during several seconds with a near infrared light source. Authors demonstrated the crucial role of superhydrophobicity of the launching disc platform for the unit takeoff and significant influence of the softness of connection between the platform and the lifter on the lifting height, stated that the lifter unit outperform a cylindrical metal coil spring in propulsion force to mass ratio. The possibility of the flight control in horizontal direction was demonstrated. It was achieved by shortening of one of the two elastic band connecting the lifter and the platform. Finally, authors have shown a possible application of the proposed device for removal of microrobots from water. The manuscript is written short and clear.

We thank the reviewer for careful review of our work. Your constructive comments have helped us to further improve our manuscript. We have addressed every comment and made corresponding changes to the manuscript and Supplementary Information. Please check the following point-by-point response for details.

My general considerations:

- 1. It is not fair to compare mechanically driven jumping insects with engineering systems using jet propulsion. It is similar to comparison of an airplane with a bird. Even a 7 m jump of dolphins *Tursiops truncatus* could be outperformed by flight of a small rocket launched even directly from the water surface.**

The authors appreciate the reviewer’s comments. First of all, we would like to clarify that the actuation mechanism of the hydrogel actuator used in the demonstrated water surface jumping soft robot originates from the instantaneous release of accumulated strain energy rather than from the gas jet providing lift. We have recorded the take-off process of the hydrogel actuator from the floated PDMS pad use high speed camera, and the results have been added to Supplementary Fig. 6 (as shown in Figure R1.1), the hydrogel actuator consists of a copolymerized hydrogel network doped with graphene. When irradiated with NIR light, deformation occurs first due to the fast phase transition (i.e., water vaporization) triggered by the photothermal response of graphene ((1)-(5) in Figure R1.1). This induces the strain energy accumulation in the hydrogel network which is characterized by high elasticity and toughness. Once the limit is reached, the accumulated strain energy

is released instantly through fracture on the bottom surface. The ultrafast deformation of the actuator's interior hits the platform to produce impulse as the actuation force that accompanied with vapour ejection ((5)-(8) in Figure R1.1). Therefore, the actuation process of the hydrogel actuator is similar to the force output by releasing a compressed spring (i.e., the long energy accumulation period through deformation and the instantaneous removal of "latch" for energy release), which is an instantaneous impulse applied to the platform with a short interaction period. It should be noted that the propulsion mechanism of hydrogel actuator is distinct from the mechanism for launching fireworks or rocket, which relies on combustion with a propellant to produce a high-temperature gas.

Figure R1.1 (Supplementary Fig. 6). The high-speed camera images showing the take-off process of the hydrogel actuator from the floated PDMS platform on water surface.

Figure R1.2 (Supplementary Fig. 10). (a) The weight difference before and after the light driven launching motion. Data points are shown as mean \pm s.d. ($n = 5$). (b) The vertical displacement of the hydrogel actuator's center of mass which actuated by the vapor jet.

The phase-change of water induces deformation and strain energy accumulation, we have measured that the average weight loss (i.e., water loss before and after the actuation) is 6.54 mg for each actuation. Such a small amount of water vapor evaporation indicates the main source of actuation for high-speed motion is not the vapour jet propulsion (as shown in Supplementary Fig. 10a, Figure R1.2a). We have roughly calculated the energy loss from the vapour jet. Specifically, as shown in Figure R1.2b, the hydrogel actuator is suspended in the air by a soft and thin wire. The vapour jetting can induce the pendulum motion of the hydrogel actuator in the air, the energy production from the vapour jetting can be roughly calculated from the increase in gravitational potential energy (U_g):

$$U_{vapour} = U_g = mgh$$

Here, m is the mass of the hydrogel actuator, h is the vertical displacement of the center of mass as shown in Figure R1.2b. The calculated results of 3.96×10^{-3} mJ is negligible compared with the kinetic energy of the robot ($E_{jumping} = 2.54$ mJ) and released energy of the hydrogel actuator ($E_{explosion} = 7.56$ mJ), respectively.

In summary, the actuation of our hydrogel actuator is similar to the reported engineered water surface jumping systems based on the latch-spring mechanism, which utilizes considerable energy accumulation and rapid release to achieve instantaneous pulse actuation (i.e., fracture in hydrogel actuator or latch remove in latch-spring systems). In addition, natural insects also rely on the energy accumulation through muscle contraction over a longer timescale and release it instantly to actuate legs or catapults (e.g., the legs of water striders and the catapults of springtails) that strike the water surface to generate instantaneous driving force. Therefore, in essence, the above-mentioned water surface jumpers employ similar actuation force production strategies. The purpose of comparison is to highlight the superiority of the proposed water surface jumping mechanism over the surface tension-dominated water surface jumping mechanism in terms of performance improvement. To better present this, we have added above discussion into Supplementary Note 3 and further added a clearer description in manuscript on Page 12, line 229 as follows:

“In this work, we adopt our previously reported hydrogel actuator⁴¹ in the fabrication of water surface jumper as a demonstration, which relies on the strain energy accumulation during deformation and instant release through fracture to realize the powerful actuation.”

2. To the disadvantages of the proposed system I would list the necessity of an external energy source to evaporate some water in the polymer block and its' single-shot usage. It shouldn't be also an imperative to remove the launching platform from water.

We thank the reviewer for making the points out. For the first concern, we would like to clarify that this work focuses on the fundamental mechanism of water surface jumping motion. The soft water surface jumping robot shown in the manuscript is a representative demonstration to help illustrate the proposed mechanism. Among others, the NIR light driven hydrogel actuator serves as a convenient and typical candidate based on our former work, which satisfy the requirement of the actuation module based on the proposed mechanism, i.e., ultrafast response time (0.6 ms) and large force-to-mass ratio ($F/m = 1.34 \times 10^4$ N/kg). For the current design, we totally agree with the reviewer's comment that the external energy source to trigger the phase transition inside the hydrogel actuator is indispensable and also induces the drawback of single time use. The reason we choose the light-driven hydrogel actuator in this work is due to the large power output (i.e., F/m ratio) and the ultrafast response speed, despite its destructive nature. Similar to the functions of soft robot demonstrated in the manuscript, we note that the light driven hydrogel actuator used is also a representative example to help understand the design principles of the actuation module. Other rigid or elastic actuation structures with faster response times and greater force-to-mass ratio can also be utilized for actuation of the soft robot.

For the launching platform, the jumping performance increases if the PDMS pad is not removed from the water. As the reviewer's suggestion, we tested the jumping performance of the hydrogel actuator from the floating PDMS platform (Supplementary Fig. 17a and 17b, as shown in Figure R1.3a and R1.3b) without removing launching platform from the water. The hydrogel actuator is launched with high initial velocity and the launching height can reach to up to 1.7 m (Figure R1.3c). During the actuation process, the ultrafast deformation and powerful force output of the hydrogel actuator hit the platform, acquiring the reaction force for the jumping motion. The platform is pressed into the water and remains there due to water surface broken. The hydrogel actuator is finally separated from the platform and exhibits the high-performance jumping behavior. In this work, we take the launch pad as a part of a water surface jumping robot which require all components leaving from water to accomplish a water surface jumping motion.

Figure R1.3 (Supplementary Fig. 17). (a) Schematic illustration and (b) image of the initial state for jumping motion of the hydrogel actuator without the remove the launching pad from water. (c) The overlaid image shows the launching height of the hydrogel actuator.

We have edited the discussion on page 11, line 199 using the above-mentioned figure shown as follows:

“In other word, the momentum transfer strategy enabled by the soft connection allows for a driving force that exceeds the threshold required to break the water surface, enabling the actuator to maximize its output and achieve a higher initial velocity without concern for surface disruption (Fig. 3b). Although the jumping motion without the removing process of launching platform from water can achieve more attractive motion performance (Supplementary Fig. 17), in this work, we still take the launch pad as a part of a water surface jumping robot which required all components leaving from water to accomplish a water surface jumping motion.”

3. The lifting height of the unit is indeed primary depends on drag and buoyant forces and only partially depends on surface tension, but its’ stationary floating on the water surface still depends on surface tension forces, since the platform should be superhydrophobic for the successful unit flight.

The authors appreciate the reviewer’s comments and totally agree that the initial floating state on the water surface primarily relies on the surface tension. Similar to insects and engineered robotics

with water surface jumping motion abilities that utilize surface tension for floating, the initial stable state on the water surface is indispensable for the proposed water surface jumping mechanism and current demonstrated soft robot. The stable floating state on the water surface ensures the energy storage and release process that necessary for water surface jumping motion. We fabricated the superhydrophobic structure on the launching pad to increase the surface tension, producing a more stable initial floating state that supports the actuation module while providing solid interaction for initial momentum acquisition.

We have edited the manuscript and added discussion on page 14, line 257 as follows,

“The third design principle involves the floating method of the water surface jumper. The initial stable floating state is indispensable that relies on the surface tension of water. In this work, we used PDMS as a hydrophobic material to fabricate the floating pad that supports the actuation module while providing solid interaction for initial momentum acquisition. Additionally, we fabricated micropillar arrays (diameter = 90 μm , height = 200 μm) on the PDMS pad to enhance its superhydrophobicity (Fig. 3e and Supplementary Fig. 23)”

4. According to Fig. 3d, the propulsion force of the proposed unit should be $5 \times 10^4 \times 0.4 \times 10^{-3} = 20$ N. This considerably differs from the highest value presented in the manuscript (e.g., Fig. 1, Tab. 1). Besides, even if acceleration, $F/m = 1.34 \times 10^4$ (Tab. 1), was acting during 0.6 ms (Line 97), then the take-off velocity should be 8.25 m/s. That was obviously not the case.

(1) We thank the reviewer for careful review and for pointing this out. We sincerely apologize for the misunderstanding caused by the lack of clear remarks of the weight of the single actuation module ($m_{actuator}$) and whole systems (m_{total}). Based on the proposed water surface jumping mechanism, the powerful actuation module with fast-response characteristic and large driving force (F) to generate high acceleration and initial velocity is indispensable, thereby improving the jumping performance. In addition, the total body mass of the jumper is strictly limited by the bearing capacity of water surface, necessitating an actuator with a high output driving force-to-body mass ratio ($F/m_{actuator}$) for greater acceleration. For the data shown in Fig. 3c and 3d, the $F/m_{actuator}$ ratio is calculated for individual actuation module (i.e., hydrogel actuator and spring actuators) rather than the entire robot system for the comparison, the force measurement of the hydrogel actuator are as follows, the hydrogel actuator is placed on the surface of the force sensor (Nano17ti: acquisition rate is 33.3 kHz) and triggered by NIR light, the hydrogel actuator hits the force sensor and the instantaneous force data are recorded. Therefore, the F/m ratio of the hydrogel actuator can be calculated based on the $F/m_{actuator} = 5.39 \text{ N} / 110 \text{ mg} = 4.90 \times 10^4 \text{ N/kg}$ (as shown in Fig. 3d), which

is consistent with Fig. 1. The comparison of the $F/m_{actuator}$ is to highlight the design principle of the actuation module.

In addition, for the data shown in the Table 1 and Supplementary Note 1, i.e., $F_{driving}/m_{total} = 5.5 \text{ N} / 410 \text{ mg} = 1.34 \times 10^4 \text{ N/kg}$, is calculated based on the detected maximum force (i.e., 5.5 N) (as shown in Supplementary Fig. 21a) by above mentioned force measurement method and the weight of whole robot system (i.e., the total weight (410 mg) of actuation module, launching pad and soft tendons). The reason for using the robot's mass for calculation is that our comparison objects include natural insects (e.g., water strider and springtails), whose force output requires the body's skeletal muscle system to cooperate rather than the operation by single catapult organs, and it is difficult for us to find the weight data of these catapult organs. To ensure the fairness of the comparison, we use the overall mass of the soft robot and animals to calculate the F/m_{total} for comparison.

In the manuscript, we have edited the Table 1 and Fig. 3d with specific expressions of F/m_{total} and $F/m_{actuator}$, respectively, and added the notes in the caption shown as follows:

		Superhydrophobic contact with water	Actuation time	F/m_{total} (N/kg)	Allow water surface broken
Surface tension-dominated mechanism	Animals (Springtails (2))	√	$< 10^0 \text{ ms}$ (0.5 ms) (2)	2.92×10^3 (2)	×
	Artificial (Water strider bionic robot) (1)	√	$10^0 - 10^1 \text{ ms}$ (15 ms) (1)	1.36×10^2 (1)	×
This work		√	$< 10^0 \text{ ms}$	1.34×10^4	√

Table R1.1 (Table 1). The comparison of the previously reported surface tension-dominated mechanism employed by animals and artificial jumpers and our proposed water surface jumping mechanism. Note the detailed calculation of F/m_{total} can be found in Supplementary Note 1.

Figure R1.4 (Fig. 3d). (d)The comparison of the output force/mass ratio and response time of the representative actuators. Note the detailed calculation of $F/m_{actuator}$ can be found in Supplementary Note 1.

In addition, we added the Table R1.2 (Supplementary Table 3) and corresponding descriptions in Supplementary Note 1 to clarify the force measurement methods and the data used in Fig. 3d, shown as follows:

“In addition, the total body mass of the jumper is strictly limited by the bearing capacity of water surface in the initial stable float state, necessitating an actuator with a high output driving force-to-body mass ratio ($F/m_{actuator}$) for greater acceleration. For the data shown in Fig. 3c and 3d, the $F/m_{actuator}$ ratio is calculated for individual actuation module (i.e., hydrogel actuator and spring actuators) rather than the entire robot system for the comparison, the force measurement of the hydrogel actuator are as follows, the hydrogel actuator is placed on the surface of the force sensor (Nano17ti: acquisition rate is 33.3 kHz) and triggered by NIR light, the hydrogel actuator hits the force sensor and the instantaneous force data are recorded. For the spring actuators, the maximum force was detected during the extreme compression as illustrated in the inset of Fig. 3d and Supplementary Fig. 21. As shown in the Supplementary Table 3, the results show that the hydrogel actuator significantly outperforms spring actuators in terms of the driving force-to-mass ratio and light body.

	Driving force (N)	Body mass (mg)	$F/m_{actuator}$ (N/kg)
Soft spring actuator	6.07	197	3.08×10^4
Hard spring actuator	10.80	356	3.04×10^4
Hydrogel actuator	5.39	110	4.90×10^4

Table R1.2 (Supplementary Table 3). The detailed calculation parameters and comparison of the soft spring actuator, hard spring actuator and our hydrogel actuator.

(2) For the calculation of the take-off velocity, when the actuation process begins, the hydrogel actuator is triggered and hits the launching pad, the actuation module and the superhydrophobic pad experience equal and opposite forces, resulting in the upward movement of the actuation module and the downward movement of the superhydrophobic pad. The initial upward motion of the actuation module encounters only air resistance and gravity, and the superhydrophobic pad moves downward into the water, facing multiple resistances, including hydrostatic force, bounce force and surface tension. Until the tendons straighten that induced the velocity rematch (i.e., take-off velocity) between the actuation module and the superhydrophobic pad. In this process, energy loss occurs

in the process of energy conduction and conversion from the hydrogel actuator, overcoming the dynamic interaction with water, and water resistance when pulling out the pad. Therefore, the take-off velocity is not suitable calculated by $V = F/m \cdot t$, which does not consider the energy losing during the actuation process.

We calculated the take-off velocity by modeling the post-take-off motion of the jumper as deceleration due to the gravity and the air drag force ($F_{drag} = C_d \rho \pi r_0^2 v_f^2 / 2$, where v_f is the flying velocity),⁵

$$v_{take-off} = \sqrt{\frac{\left[e^{\frac{h C_d \rho \pi r_0^2}{m}} - 1 \right] \times 2mg}{C_d \rho \pi r_0^2}}$$

Here, C_d is the drag coefficient, h is the maximum jumping height, ρ is the density of air, r_0 is the basal radius of the soft robot, m is the total mass of the jumper. For the recorded highest water surface jumping motion, the calculated take-off velocity is 3.52 m/s.

In addition, as shown in the Fig. 1c-(II), the single hydrogel actuator is launched on a floating PDMS launching pad. The hydrogel actuator is launched at a very high speed, at the same time, the launching pad is pressed into the water. The initial speed recorded by high-speed camera is 7.94 m/s, which shows limited errors compared to the reviewer's calculation (i.e., 8.25 m/s). The reason is that the instantaneous solid interaction between hydrogel actuator and PDMS pad only experience limited energy loss.

5. Not self-consistent results regarding jumping speed: Fig. 1e, Fig. 1d(III), Fig. 2, Fig. 3b, Fig. 4e, Supplementary Tab. 1; and height: Fig. 2, Fig. 4d, Fig. 4f, Supplementary Tab. 1, Supplementary Fig. 8a.

We thank the reviewer for careful review and making this point out. We clarify that all the data are calculated or read based on the Videos have been provided in the Manuscript or Supplementary Information. We acknowledge that the performance of each water jumping event has a certain deviation, which is related to many factors. For example, the light driven method as a remote wireless strategy that the incident light is easily affected by the scattering and reflection of the surface of the robot. In addition, the slight deviations light incident angle also cause difference in trigger process, thus inducing the deviations in jumping performance. Moreover, the explosive actuation method of the hydrogel actuator is a violent form of energy release. It is hard for us to control the energy and force output to be same for each jumping motion, which also leads to deviations in its jumping performance.

For the jumping speed shown in Fig. 1e (we guess that the reviewer’s comment is referring to Fig. 1f), Fig. 1d (III), Fig. 2, Fig. 3b, and Supplementary Tab 1, are calculated based on the Supplementary Video 1 and Supplementary Video 2 that have been provided. We clarify that the $V_b = 4.59$ m/s shown in Fig. 1d (III) and Fig. 1f is the speed of the actuation module, which is higher than the total speed (i.e., 3.52 m/s) due to the different motion direction of the actuation module and launching pad. For the speed (i.e., take-off velocity) shown in Fig. 1d (III) (i.e., $V_{total} = 3.52$ m/s), Fig. 2, Fig. 3b, and Supplementary Tab 1 is the calculated uniform speed of whole soft robot. These data are consisting well with negligible deviations. **To avoid misunderstanding of these two speeds, we added the note in the captions of these figures shown as follows,**

“Fig. 1. Key design principles of developed water surface jumping mechanism. (a) Schematic illustration of the functional parts of the springtails and (b) its water surface jumping motion based on the surface tension-dominated mechanism. (c) Schematic illustration the functional modules of the water surface jumping soft robot based on proposed design principles and (d) the high-speed camera images obtained from Supplementary Video 1 showing the take-off process of the soft robot. **Note that the V_b and V_{total} in the image is the speed of the actuation module and the take-off velocity of the whole soft robot, respectively.** (e) The comparison of the robot with and without the superhydrophobic contact with water. (f) The comparison of the robot with soft connection and rigid connection.”

“Fig. 2. Performance of the demonstrated water surface jumper based on proposed mechanism. (a) Radar chart illustrates the performance comparison for the water surface jumper based on the proposed mechanism with natural and engineered water surface jumping system. (b) Jumping height and take-off velocity comparison of the soft robot in the current study with previously reported engineered water surface jumping robots and insects^{1,2,10,11,22,25,27,34-37}. **Note that the take-off speed shown in Fig. 2 are obtained from Supplementary Video 2.**”

“Fig. 3. Comparison and analysis of three design principles of the proposed mechanism. (a) Schematic illustration the initial momentum acquisition of the soft and rigid energy transfer system. (b) The take-off velocity change of the soft and rigid connection robots. **Note that the take-off velocity is the whole speed of the soft robot that obtained from Supplementary Video 1.** (c) The high speed camera images illustrate the response time of the representative actuators (i.e., hard spring, soft spring and fast-response actuator) launching from glass substrate. Note that the initial stresses are set to approximately 5.5 N. (d) The comparison of the output force/mass ratio and response time of the representative actuators launching from glass substrate. Note the detailed calculation of $F/m_{actuator}$ can be found in Supplementary Note 1. (e) The image illustration of the experimental setup, the PDMS pad with and without micropillar array mounted on a force sensor and slowly pulled out from the water surface at a constant speed of 0.1 mm/s (left). The contact angle of the PDMS pad with and without micropillar array (right). Inset: SEM image of the top view of the micropillar array. (f) The surface tension change during the process of the PDMS pad pulled out from water surface.”

For the speed shown in Fig. 4e which was obtained from the Supplementary Video 4, we clarify that the light intensity in Fig. 4 is 2.8 W (4.4 W/cm²) due to the limited height of grid background board and to avoid the larger deviations caused by higher light intensity. For the current soft robot demonstrated in the manuscript, the jumping performance can be influenced by many factors, e.g., light intensity, power output of the hydrogel actuator, hinge length and diameter of the launching pad, etc. We have tested these influence factors and the results have been added as Supplementary Note 5 in the Supplementary information. Among others, the jumping height of the soft robot is influenced by light intensity as shown in Supplementary Fig. 14a (Figure R1.5). The change in the light energy input causes the difference in thermal production and phase transition volume. The increased light intensity means faster thermal production and deeper penetration, which induce the bigger phase transition volume inside the hydrogel actuator and more vapor accumulation before fracture, thus increasing the force output of the hydrogel actuator and the jumping performance of the robot.”

Figure R1.5 (Supplementary Fig. 14). The influence factors of the jumping performance of the soft robot. (a) The jumping height and response time of water surface jumping motion for soft robot irradiated with different light intensity. (b) The jumping height of the water surface jumping motion for soft robot with different graphene content. (c) Schematic illustration of the graphene content induced the difference of phase transition volume inside the actuator's body. (d) The jumping height of water surface jumping motion for soft robot and stress-strain curve of hydrogel with different (d, e) water content and (f, g) MBA content, respectively. (h) The jumping height and contact area with different diameters of pad. (i) The jumping height of the soft robot with different (i) softness and (j) length of the tendon. Note that the data are captured with the light intensity is 2.8 W ($4.4 W/cm^2$).

We revised the manuscript on page 14, line 275 shown as follows,

“For the current design, the jumping performance of the soft robot can be influenced by many factors, e.g., light intensity, power output of the hydrogel actuator, hinge length and diameter of the launching pad, etc. We have tested these influence factors and the detailed discussion can be

found in Supplementary Note 5.”

We further added the note to the caption of Fig. 4 on page 16, line 289 shown as follows,

“Fig. 4. Motion control of the water surface jumping soft robot. (a) Schematic illustration showing the direction control principle of the water surface jumping soft robot. (b) Overlaid images obtained from Supplementary Video 3 showing the directional water surface jumping motions. (c) High-speed camera images showing the directional take-off process. Note that the tendon length of the left : right is 3 : 1.5 (cm). (d) The jumping trajectories of the soft robots with different left/right tendon length ratios. (e) The take-off velocity and take-off angle of the soft robots with different left/right tendon length ratios. (f) The jumping distance and jumping height of the soft robots with different left/right tendon length ratios. (g, h) The overlaid images indicating the soft robot with representative left/right tendon length ratio across railings with different heights.

Note that all the data captured in Fig. 4 are with the light intensity of 2.8 W (4.4 W/cm²).”

(2) The jumping height of the soft robot shown in Fig. 2b, Supplementary Tab. 1, Supplementary Fig. 8a was obtained from Supplementary Video 2 with the light intensity of 6.3 W (9.91 W/cm²). We checked Supplementary Video 4 and recalculated the speed and jumping height, the revised Fig. 4 is shown as follows. Note that the jumping trajectory and velocity are calculated based on the taken videos that the grid background board as reference. Although the data presented may be biased due to slight deviations in the centroid position read, we still believe that it represents an accurate trend of the soft tendon length ratio.

Further comments:

6. No light source power is given.

The authors appreciate the reviewer for making this point out. We have edited the description on page 27, line 503, as follows:

“In the current study, unless otherwise mentioned, the water content, graphene content and light intensity were 60 wt%, 0.17 wt% and 6.3 W (9.91 W/cm²), respectively.”

We have added the light source power in the caption of Fig. 4d - 4f on page 16, line 289 as follows:

“Fig. 4. Motion control of the water surface jumping soft robot. (a) Schematic illustration showing the direction control principle of the water surface jumping soft robot. (b) Overlaid images

obtained from Supplementary Video 3 showing the directional water surface jumping motions. (c) High-speed camera images showing the directional take-off process. Note that the tendon length of the left : right is 3 : 1.5 (cm). (d) The jumping trajectories of the soft robots with different left/right tendon length ratios. (e) The take-off velocity and take-off angle of the soft robots with different left/right tendon length ratios. (f) The jumping distance and jumping height of the soft robots with different left/right tendon length ratios. (g, h) The overlaid images indicating the soft robot with representative left/right tendon length ratio across railings with different heights. **Note that all the data captured in Fig. 4 are with the light intensity of 2.8 W (4.4 W/cm²).**”

7. Page 3, Line 57: “0.5 ms”, the muscle contraction is never so fast. Just read the reference you provided carefully.

The authors appreciate the reviewer for making this point out and agree that the inappropriate words of “muscle strokes” were used. What we want to express is the ejection time of the springtails catapult (i.e., furcula). This process is the instantaneous muscle-controlled activation of latch removal (energy release) after the energy accumulation through muscles contraction to realize the power amplification. **We have replaced the words of “muscle stroke” by “muscle activation” in the whole manuscript shown as follows,**

On page 2, line 35, “a light-weight, high-power actuation module capable of providing significant propulsion force within an ultrashort time (comparable to muscle activation timescales) for effective initial momentum acquisition”

On page 3, line 58, “Energetics and kinematics: high momentum transfer efficiency is achieved through ultrafast muscle activations (e.g., approximately < 1 ms for springtails), sophisticated momentum transmission system and careful adjustment of propulsion force to remain just below the threshold needed to break the water surface.

On page 4, line 82, “This inefficiency arises from the long actuation duration (typically is $\sim 10^1$ milliseconds) caused by the relatively unsophisticated design of actuation and transmission mechanisms compared to some of the nature species (e.g., springtails) with high-performance motion behavior, which rely on the fast muscle activation ($\sim 10^0$ millisecond)”

On page 5, line 99, “Light-weight, powerful actuation module: capable of providing adequate propulsion force in an ultrashort time (comparable to rapid muscle activation in semi-aquatic arthropods), this allows the jumper to acquire momentum within a short time window (e.g., 0.6 ms in Fig. 1d-II).”

On page 20, line 367, “a light-weight powerful actuation module that can provide adequate propulsion force in an ultrashort time (muscle activation timescale) for large initial momentum acquisition, and a soft momentum transmission system for highly efficient kinetic energy transfer.”

7. Page 4, Line 81: “... the long actuation duration ($\sim 10^1$ milliseconds)...”, however, actuation time for Motor-spring is 2 ms, according to Supplementary Tab. 1.

The authors appreciate the reviewer’s comment. The definition of the take-off time and actuation time shown in the Supplementary Table 1 is the “period from actuator start response to the robot’s body leaving the water surface” and the “impulse time of the actuator” (i.e., the time period during which the outputted force of actuator generates an impulse on the robot), respectively, (as shown in Figure R1.6). From the reference,⁶ we found the data of the impulse time that conduct by mathematically prediction is 2 ms. However, we didn’t find the take-off time and cannot estimate by the data given in the article. In addition, from the limited high speed camera images that shown in the article, we can infer that its take-off time should be close to or greater than 10 ms, because the high-speed images of the water surface jumping motion shows that it takes 4.5 ms from the splash after the robot hitting the water surface to completely detached with water surface (Fig. 10 shown in the reference). Based on the impulse actuation and complex transmission structure design of this robot, we believe that the soft robot we demonstrated has a shorter take-off time period (i.e., 6 ms shown in Fig. 1d-(III) in the manuscript). For accuracy, we have revised the take-off time and actuation time of this reference in the Supplementary Table 1 to N/A and 2 ms respectively. We further edited the description on page 4, line 82 shown as follows,

“This inefficiency arises from the long take-off period (typically is $\sim 10^1$ milliseconds) caused by the relatively unsophisticated design of actuation and transmission mechanisms compared to some of the nature species (e.g., springtails) with high-performance motion behavior, which rely on the fast muscle activations ($\sim 10^0$ millisecond).”

Figure R1.6. The definition of the (a) actuation time and (b) take-off time.

8. Page 4, Lines 82-83: “... fast muscle strokes ($\sim 10^0$ millisecond)...” , however, actuation time for e.g. fisher spider is 52 ms, according to Supplementary Tab. 1.

We thank the reviewer for careful review and making this point out. Some water surface-jumping insects with larger body size, especially those that rely on their limbs for jumping, require longer take-off time period, and some insects with shorter take-off times (e.g., 1.5 ms of springtails) exhibit motion performance far superior to theirs (as shown in Fig. 2b). We have edited the description on page 4, line 82 shown as follows,

“This inefficiency arises from the long take-off period (typically is $\sim 10^1$ milliseconds) caused by the relatively unsophisticated design of actuation and transmission mechanisms compared to some of the nature species (e.g., springtails) with high-performance motion behavior, which rely on the fast muscle activations ($\sim 10^0$ millisecond).”

9. Page 5, Lines 99-100: “... due to the incompressibility of water (Fig. 1e).”, water is incompressible in both situations presented in top and bottom images. Buoyant force, conventional and wave drag forces propose the required for the lift up reaction force.

The authors appreciate the reviewer’s careful review and comment. As the reviewer’s suggestion, we have edited the description on page 5, line 100 shown as follows:

“During this brief period, the water beneath the launching pad experiences limited deformation, allowing for a bounce force greater than the maximum reaction force that surface tension can provide, due to the incompressibility of water and hydro-dynamic interaction (Fig. 1e).”

10. Page 5, Lines 104-105: “... rigid connections would result in momentum mismatch and hinder the initial velocity acquisition...”, the actual mechanism is that the stem/water stream coming out of the lifter hits the platform. And momentum of the stream is transferred to the platform, which is almost equal and opposite to that of the lifter.

The authors appreciate the reviewer’s comments. We would like to clarify that the actuation mechanism of the hydrogel actuator used in the demonstrated water surface jumping soft robot originates from the instantaneous release of accumulated strain energy rather than from the gas jet providing lift. We recorded the take-off process of the hydrogel actuator from the floating PDMS platform using a high speed camera as shown in Supplementary Fig. 6 (Figure R1.7), the hydrogel actuator consists of a copolymerized hydrogel network doped with graphene. When irradiated with NIR light, deformation occurs first due to the fast phase transition (i.e., water vaporization) triggered by the photothermal response of graphene ((1)-(5) in Figure R1.7). This causes strain energy accumulated in the hydrogel network which possesses high elasticity and toughness. Once the limit is reached, the ultrafast deformation of the actuator’s interior hits the platform to produce impulse as the actuation force that accompanied with vapour ejection ((5)-(8) in Figure R1.7). Therefore, the actuation process is similar to the force output by releasing a compressed spring, which is an instantaneous pulse applied to the platform with a short interaction period. It should be noted that the propulsion mechanism demonstrated of hydrogel actuator is distinct from the mechanism for launching fireworks or rocket, which relies on combustion with a propellant to produce a high-temperature gas.

Figure R1.7 (Supplementary Fig. 6). The high-speed camera images showing the take-off process of the hydrogel actuator from the floated PDMS platform on water surface.

Figure R1.8 (Supplementary Fig. 10). (a) The weight different before and after the light driven launching motion. Data points are shown as mean \pm s.d. ($n = 5$). (b) The vertical displacement of the hydrogel actuator's center of mass which actuated by the vapor jet.

The phase-change of water induces the deformation and strain energy accumulation, we have measured the average weight loss (i.e. water loss before and after the actuation) is 6.54 mg for each actuation, such a small amount of water vapor evaporation indicates the main source of actuation for high-speed motion is not the jet propulsion (as shown in Supplementary Fig. 10, Figure R1.8a). We have roughly calculated the energy loss from the vapor jet. Specifically, as shown in Figure R1.8b, the hydrogel actuator is suspended in the air by a soft and thin wire, the vapour jetting can induce the pendulum motion of the hydrogel actuator in air, the energy production from the vapour jetting can be roughly calculated from the increase in the gravitational potential energy (U_g):

$$U_{vapour} = U_g = mgh$$

Here m is the mass of the hydrogel actuator, h is the vertical displacement of the center of mass as shown in Figure R1.8b, the calculated results of 3.96×10^{-3} mJ is negligible compared with the kinetic energy of the robot ($E_{jumping} = 2.54$ mJ) and the released energy of the hydrogel actuator ($E_{explosion} = 7.56$ mJ), respectively.

Therefore, the actuation force mainly originates from the reaction force that produced by hydrogel actuator hitting the launching pad. When the system is fabricated by rigid connection (as shown in Fig. 1f-bottom), the robot can be considered as a whole. The actuation force generated from the actuator makes the robot press the water surface downward, which is similar to the initial actuation state of the reported surface tension dominated systems. However, during the short period of acceleration, the robot is subject to the dynamic resistance of the water due to the water surface breaking

and a large amount of energy loss caused by the splash of water, resulting in the robot obtaining a small initial velocity or being unable to detach from the water surface.

We have revised Fig. 1 and added the calculation process and discussion of the energy loss from vapour jetting to Supplementary Note 3.

11. Fig. 2b: Water strider should be instead of “Water spider”.

The authors sincerely appreciate the reviewer’s careful review and making this point out. We have revised the Fig. 2b shown as follows,

We further edited the description on page 3, line 51 shown as follows,

“(I) Surface tension utilization: semi-aquatic arthropods possess superhydrophobic bodies (e.g., the hairy, superhydrophobic legs of water strider) and low body mass, allowing them to remain on the water surface without sinking by exploiting the surface tension of water.”

12. Fig. 3.: Figures should appear after they first mentioned. c and d, it should be noticed, if the results are obtained by launching the actuators from hard substrate and not from the water surface.

We thank the reviewer for making this point out. As the reviewer’s suggestion, we have added the description on the figure captions shown as follows,

“Fig. 3. Comparison and analysis of three design principles of the proposed mechanism. (a) Schematic illustration the initial momentum acquisition of the soft and rigid energy transfer system. (b) The take-off velocity change of the soft and rigid connection robots. Note that the take-off velocity is the whole speed of the soft robot that obtained from Supplementary Video 1. (c) The

high speed camera images illustrate the response time of the representative actuators (i.e., hard spring, soft spring and fast-response actuator) launching from glass substrate. Note that the initial stresses are set to approximately 5.5 N. (d) The comparison of the output force/mass ratio and response time of the representative actuators launching from glass substrate. Note the detailed calculation of $F/m_{actuator}$ can be found in Supplementary Note 1. (e) The image illustration of the experimental setup, the PDMS pad with and without micropillar array mounted on a force sensor and slowly pulled out from the water surface at a constant speed of 0.1 mm/s (left). The contact angle of the PDMS pad with and without micropillar array (right). Inset: SEM image of the top view of the micropillar array. (f) The surface tension change during the process of the PDMS pad pulled out from water surface.”

13. Eq. 1: Both m_b and m_p or their ratio should be given.

The authors appreciate the reviewer’s comments. As the reviewer’s suggestion, we have added the value of \$m_b\$ and \$m_p\$ on page 11, line 194 shown as follows:

$$v_{total} = \frac{m_b v_b + m_p v_p}{m_b + m_p} \quad (1)$$

where m_b (266 mg), v_b , m_p (144 mg), and v_p are the masses and velocities of the robot’s body and superhydrophobic pad, respectively. The larger driving force from the actuator results in higher velocity of the body (v_b) (i.e., actuation module) and a more rapid decay of the superhydrophobic pad’s downward velocity (v_p) until it approaches zero.”

14. Page 11, Lines 185-186: “This increased velocity difference between v_b and v_p results in a more pronounced upward movement of whole jumper...”, the more the velocities difference, the less the total velocity, according to Eq. 1.

The authors appreciate the reviewer’s comment. When the actuation process begins, the actuation module and the superhydrophobic pad experience equal and opposite forces, resulting in the upward movement of the actuation module and the downward movement of the superhydrophobic pad. Because of the soft tendon connection (i.e., no drag force), the initial upward motion of the actuation module encounters only air resistance and gravity, allowing it to achieve a high initial velocity with minimal reduction. In contrast, the superhydrophobic pad moves downward into the water, facing multiple resistances, including hydrostatic force, bounce force and surface tension.

These resistances lead to a more rapid velocity reduction for the superhydrophobic pad in water, until the tendons straighten that induced the velocity of superhydrophobic pad reach to zero. During this process, the mismatch of the motion velocity decrease rate induces the greater velocity difference, which means that the bigger upward velocity of the actuation module when the pad's velocity decreased to zero, thus inducing the higher take-off velocity.

We have edited the manuscript on page 11, line 195 shown as follows,

“The larger driving force from the actuator results in higher upward velocity of the body (v_b) (i.e., actuation module) and a more rapid decay of the superhydrophobic pad's downward velocity (v_p) until it approaches zero. The difference in the velocity decay rate in opposite directions between v_b and v_p results in an upward movement and take-off velocity of whole jumper (as described by equation (1)).”

15. Page 11, Lines 199-201: “The released energy by the hydrogel actuator applied to the floating pad and the jumping energy utilization efficiency are estimated to be 3.97 mJ and 63.97 %, respectively (see calculation details in Supplementary Note 3).”, based on the size of the lifter in Supplementary Fig. 6 and Eq. 7, E_b should be ~100 times larger.

We thank the reviewer for making this point out and agree that the calculated result is less than the actual value due to the kinetic energy of the water and the potential energy of the splashed water are ignored. We have recalculated the energy loss during the hydrodynamic interaction of the soft robot and the jumping energy efficiency, the details have been added in Supplementary Note 3. As shown in Supplementary Fig. 8 (also shown as Figure R1.9 below), when the launching pad of the soft robot hits the water surface that induced the water surface broken and causing cavities and liquid splashing. The energy output of the actuator is converted into the kinetic energy of the robot and also the dynamic kinetic energy and potential energy of the water. For the latter, it can be calculated in two parts, i.e., the potential energy and kinetic energy of water displaced beneath the water surface and the splashed water above the water surface, respectively. We analyzed a series of high-speed images and choose the critical frames for calculating, which is the highest position of the splashing water column and the lowest position of the displaced water. At this moment, the kinetic energy is zero and assuming that have been fully converted to the potential energy of the water.

Figure R1.9 (Supplementary Fig. 8). A series of images obtained from Supplementary Video 1 indicating the moment of the kinetic energy is zero.

Therefore, as mentioned above, the energy loss during the dynamic interaction with water (E_b) can be calculated by the maximum potential energy (i.e., kinetic energy is 0) of the displaced water (E_1) and splashed water (E_2) (Supplementary Fig. 9, as shown in Figure R1.10a).

$$E_b = E_1 + E_2 \quad (7)$$

For the calculation of E_1 that can be calculated through the high-speed image when the displaced water reaches to the maximum volume (Figure R1.10b):

$$E_1 = \int_0^{f(x)} F_b dx = \int_0^{f(x)} \rho g S dx \quad (8)$$

Where ρ is the density of water, g is the acceleration of gravity, S is the area of the PDMS pad, the $f(x)$ can be identified through high-speed image. In order to simplify the calculation, we use the potential energy of the displaced water as an approximation. The volume of the displaced water can be regarded as a hemisphere that the parameter as shown in Figure R1.10b, the weight of the displaced water is ~ 10 g can be calculated as follows,

$$m = \rho V_1 = \rho \cdot \frac{\pi}{3} (3r - h_1) \cdot h_1^2 \quad (9)$$

Where ρ is the density of water, r is the radius of the hemisphere, h_1 is the distance between the water surface with the lowest position of the cavity.

Therefore, the energy can be roughly estimated shown as follows,

$$E_1 = \int_0^{f(x)} F_b dx \approx m_1 g h_1 \quad (10)$$

The E_I can be calculated is 1.7 mJ.

Similarly, as shown in Figure R1.10c, the volume of the splashing water column can be roughly estimated by considering it as a hollow truncated cone, the thickness of the truncated cone can be estimated by the transparency distribution in high-speed camera screenshots, the weight of the splashed water is ~ 14.54 g can be calculated as follows,

$$m_2 = \rho(V_{outer} - V_{inner}) \quad (11)$$

Where ρ is the density of water, V_{outer} and V_{inner} is the volume of the outer and inner truncated cone, respectively, that can be calculated by $V = \frac{1}{3}\pi h(R^2 + Rr + r^2)$ based on the parameters given in Figure R1.10c.

Therefore, the potential energy of the splashed water above the water surface is 3.19 mJ can be roughly estimated by,

$$E_2 = m_2gh_2 \quad (12)$$

The energy loss during the dynamic interaction with water can be calculated is 4.89 mJ.

Figure R1.10 (Supplementary Fig. 9). (a) The energy calculation model setting and key frames. (b) The translated calculation model parameters of the maximum potential energy of the displaced water (E_I) and splashed water (E_2).

We have added the above calculation and revised the discussion in Supplementary Note 3, and we further recalculated the released energy applied on the PDMS pad of the hydrogel actuator for the jumping actuation,

$$E_{explosion} = E_s + E_b + E_{jumping} \quad (15)$$

Based on this, the released energy can be calculated through equation (13) is 7.56 mJ. And the energy conversion efficiency can be calculated through equation (14) is 33.60 %.

$$\eta = \frac{E_{\text{jumping}}}{E_{\text{explosion}}} \quad (16)$$

In the manuscript, we revised the description in Discussion on page 12, line 215:

“The released energy by the hydrogel actuator applied to the floating pad and the jumping energy utilization efficiency are estimated to be 7.56 mJ and 33.60 %, respectively (see calculation details in Supplementary Note 3).”

16. Page 13, Line 243: “... from the water surface at a constant speed of 0.1 mm/s.”, the work of adhesion significantly depends on the pull-off speed, which is in m/s range during “jumping experiments”. Besides, the work of adhesion measured by authors is ~30 times smaller than the estimated by them explosion energy.

We thank the reviewer for making this point out and totally agree that the work of surface tension significantly depends on the pull-off speed. The measurement method of the surface tension was referenced and learned from (Chen et al., *Sci. Robot.* 2, eaao5619 (2017))⁷ and (Koh et al., *Science*, 349, 517-521 (2015))⁴. For the water surface jumping robot reported in these references, the surface tension was obtained by mounted robot on a force sensor and slowly (i.e., 0.2 mm/s) pulling it out of water controlled by tensile test machine. We agree that increasing the pulling speed to the actual velocity level of the jumping motion would yield more accurate results, but it is difficult for us to find such instrument with high speed (m/s level). In addition, faster pulling speeds and the tiny surface tension require higher sampling frequency and accuracy of the force sensor.

As the reviewer’s comment, the faster pulling speed tends to result in lower measured surface tension work (i.e., <0.13 mJ that tested in quasi-static conditions), which introduces limited errors in the calculation of energy conversion efficiency compared to the released energy of the actuator (i.e., 3.97 mJ), Therefore, we use quasi-static test conditions to roughly estimate the work of surface tension.

We have added a note in the caption of Supplementary Fig. 7 shown as follows,

“**Supplementary Fig. 7.** (a) The image of the experimental setup. The robot is mounted on the force sensor and slowly pulled out from the water surface. Note that the pulling speed affects the measured surface tension. Due to the limitations of the test equipment, a slow pulling speed (0.1 mm/s) is used to roughly estimate the surface tension. (b) Force trace as the robot is pulled out of the water surface.”

In addition, we have further edited the corresponding descriptions in Supplementary Note 3 shown as follows,

“Where a is the water surface level (set as 0) in the initial state and b is the height when the PDMS pad out of the water surface, F_s is the surface tension before breaking the water surface, the E_s can be calculated from the equation (6) and Supplementary Fig. 7b is 0.13 mJ. Note that the work of surface tension is significantly influenced by the pull-off speed, the actual motion velocity is at the m/s level. Due to the limitations of the test equipment, a quasi-static measurement with slow pulling speed (0.1 mm/s) is used to roughly estimate the surface tension. The faster pulling speed tends to result in lower measured surface tension work (i.e., <0.13 mJ), which introduces limited errors in the calculation of energy conversion efficiency compared to the released energy of the actuator (i.e., 7.56 mJ as shown in the end of Supplementary Note 3). Therefore, we use quasi-static test conditions to roughly estimate the work of surface tension.

17. Page 25, Lines 451: Any black dye could be used in the jumper instead of graphene, I suppose.

The authors appreciate the reviewer’s comment and agree that the other materials with photo-thermal effects can be used for doping in the hydrogel actuator. The key point of the hydrogel actuator is that the material system has a combination of high toughness, elasticity, and rapid phase transition so that it can accumulate appreciable energy induced by the rapid phase-change ability and release stored energy instantly when elastic fracture occurs. *As the reviewer suggested, we have fabricated the hydrogel actuator by replacing the graphene with other materials (i.e., Fe₂O₃ nano particles and black ink used in the pen as shown in Supplementary Fig. 29a and 29b (Figure R1.11)).* These materials were doped by simply mixing them with the hydrogel precursor and followed by irradiating with UV light to induce the polymerization. Note that a transparent layer of hydrogel is fabricated on the top as constraint to ensure the phase change volume occurs close to the bottom surface, thus inducing the upward launching motion of the hydrogel actuator. As shown in Figure R1.11, the hydrogel actuator doped with Fe₂O₃ nanoparticles and black ink are launched from the substrate that actuated by NIR light, proves the graphene can be replaced by materials with photo-thermal abilities. Note that the performance of the actuators (take-off velocity, launching height, etc.) shown in Figure R1.11 are far away compared with the actuators demonstrated in the manuscript, which is influenced by many factors, e.g., content of doped materials, strength of the photo-thermal effect, thickness and the ratio of the different layer, etc. We only conducted feasibility experiments without further exploration as this was beyond the scope of this study.

Figure R1.11 (Supplementary Fig. 29). (a) The images and overlaid motion images of the actuators fabricated by doping (a) the Fe_2O_3 nanoparticles and (b) black ink in the hydrogel network.

We have added the description to the Method section on page 27, line 494 shown as follows,

“Note that the graphene can be replaced by other materials with photo-thermal ability (e.g., Fe_2O_3 nano particles and black ink as shown in Supplementary Fig. 29).”

18. Page 27, Lines 493: Please, specify the actual power and not the power density of the laser diode illuminating the lifter.

The authors appreciate the reviewer’s comment and added the both actual power and detected power density on the manuscript, details are shown as follows,

On page 27, line 503, “Unless otherwise mentioned, the laser power and distance between the light source and soft robot were 6.3 W (9.91 W/cm^2) and 25 cm, respectively.”

We further added the note to the caption of Fig. 4 on page 16, line 282 shown as follows,

“**Fig. 4. Motion control of the water surface jumping soft robot.** (a) Schematic illustration showing the direction control principle of the water surface jumping soft robot. (b) Overlaid images obtained from Supplementary Video 3 showing the directional water surface jumping motions. (c) High-speed camera images showing the directional take-off process. Note that the tendon length of the left : right is 3 : 1.5 (cm). (d) The

jumping trajectories of the soft robots with different left/right tendon length ratios. (e) The take-off velocity and take-off angle of the soft robots with different left/right tendon length ratios. (f) The jumping distance and jumping height of the soft robots with different left/right tendon length ratios. (g, h) The overlaid images indicating the soft robot with representative left/right tendon length ratio across railings with different heights.

Note that all the data captured in Fig. 4 are with the light intensity of 2.8 W (4.4 W/cm²).”

Reviewer #2:

This study introduces a novel water surface jumping mechanism for insect-scale robots, inspired by biological models. The design achieves jumping performance comparable to semi-aquatic arthropods by incorporating three main principles: a superhydrophobic body, a lightweight powerful actuation module, and an efficient momentum transmission system. Altogether, these design principles enabled the robot to overcome surface tension constraints, achieving effective water surface jumping. Overall, this study presents an innovative approach to insect-scale water surface jumping mechanisms, which is intriguing for the field and has potential applications in environmental monitoring or health monitoring of aquatic ecosystems through scalable data management and analysis in the cloud. Therefore, this manuscript is acceptable for this journal after addressing the following comments.

Many thanks to the reviewer for careful review and high evaluation of this work. Your constructive comments have helped us to further improve our manuscript. We have addressed each comment and made the appropriate changes in the revised version of manuscript. Please check the following point-by-point response for details.

1. The manuscript employs a hydrogel actuator, referencing the fracture-driven power amplification mechanism from “Fracture-driven power amplification in a hydrogel launcher” (Nat. Mater. 23, 1428-1435, 2024). Could the authors elaborate on any specific design or structural modifications made to the hydrogel actuator in this study compared to the one described in the 2024 Nature Materials paper? Understanding these design differences or optimizations would provide valuable insights into how this actuator has been tailored to meet the unique demands of water surface jumping in microrobotics, particularly in terms of response time, energy release, and force-to-mass ratio.

The authors appreciate the reviewer’s comment. As the reviewer’s comment, we have undertaken numerous optimizations in terms of structure design, adjustment of the material’s mechanical properties, and trigger operation compared to the paper published in Nature Materials, the detailed illustrations have been added in Supplementary Information as Supplementary Note 4. We will explain these optimizations from the following points:

(1) Structural design. As shown in Supplementary Fig. 11 (Figure R2.1), for the hydrogel launcher demonstrated in the paper published in Nature Materials, the graphene nanoparticles are evenly dispersed throughout the launcher’s body. The control of the phase-transition volume inside the hydrogel launcher is based on the light blocking effect of graphene (i.e., light incident from the

launcher's surface can only penetrate to a certain depth), the self-launching motion can only be achieved by placing the hydrogel launcher on transparent substrates. Specifically, when light is irradiated from the bottom that it can only penetrate to a limited depth, causing heat to be generated inside the hydrogel launcher closer to the bottom surface. The strain energy accumulated during the phase change induces the deformation process, which eventually triggers the explosion direction to be downward to acquire the actuation force. However, for the actuation of the water surface jumping soft robot, the water surface environment restricts the light incident from the air into the water. We solved this problem by changing the structural design to ensure that the phase change region remains inside the actuator close to the bottom surface. As shown in Figure R2.1, we adopt a two-layer structural design with a transparent hydrogel layer fabricated to allow the light to be incident from the upper surface while keeping the phase-change volume close to the bottom surface inside the actuator. In other words, it is possible to control the explosion position and force output directions based on the multi-layer strategy, i.e., the black phase-change layer and the transparent constraint layer. We hope that this can inspire the readers to do more designs of the robot based on these understandings of the mechanism.

Figure R2.1 (Supplementary Fig. 11). The structural differences of the hydrogel actuators demonstrated in this manuscript and Nature Materials.

(2) Optimization of materials and mechanical properties. The basic principle of the power amplification is long-term energy accumulation and instant release. The greater the energy accumulation and the faster release, the better power amplification and force output. Compared to the material system (i.e., P(AM-co-AA) hydrogel) of the hydrogel launcher demonstrated in Nature Materials, we have fabricated the similar copolymerized hydrogel system of P(AA-co-VP), which has a higher Young's modulus as shown in the Supplementary Fig. 12a (Figure R2.2a), indicating the higher strength of hydrogel matrix. Based on the latch-spring mechanism, the increase in the latch strength

results in more energy accumulation and faster energy release, thereby inducing better power amplification effects and greater force output.⁸ We have compared the F/m ratio of the two hydrogel actuators as shown in Figure R2.2b, the hydrogel actuator illustrated in the current manuscript has a larger F/m output value. Note that the forces are measured directly on the force sensor.

Figure R2.2 (Supplementary Fig. 12). (a) The Young's modulus and (b) F/m ratio comparison of the hydrogel actuators demonstrated in this manuscript and Nature Materials.

(3) Phase transition volume. The phase transition volume was increased by changing the spot area of NIR light irradiation. As shown in Supplementary Fig. 13a (Figure R2.3a), the larger phase transition volume achieved by increasing the light irradiation area can induce the more energy accumulation before fracture, thus leading to greater force output. i.e., increase of the F/m ratio. The water loss of the hydrogel actuator after actuation in this manuscript is higher than the launcher illustrated in Nature Materials (6.54 mg vs 3.20 mg), proving more vapour jetting induced by phase transition and more energy production for transfer and transformation. In addition, we have compared microscope images of the sectional view after actuation (Supplementary Fig. 13b, as shown in Figure R2.3b), the phase transition volume and rift of the hydrogel actuator in this manuscript is much larger than the actuator demonstrated in the paper published in Nature Materials, indicating more intense energy production and output.

Figure R2.3 (Supplementary Fig. 13). (a) The spot area and (b) phase-transition volume comparison of the hydrogel actuators demonstrated in this manuscript and Nature Materials.

The optimization of the hydrogel actuator aims to achieve faster energy release and a higher F/m ratio to meet the requirements of the actuation module for the high-performance water-jumping strategy we proposed. The focus of this manuscript is to break through the limitation of surface tension dominated mechanism to realize high-performance water surface jumping robot. The surface tension dominated water surface jumping mechanism faces an inherent physical constraint: the propulsion force must remain below the threshold required to break the water surface (i.e., avoidance of breaking the water surface). Our design strategy allows the water surface to be broken due that the actuation force originates from the solid interaction rather than the reaction force from the water surface. Each module in our design (i.e., hydrogel actuator, superhydrophobic pad, soft tendons) serves as a representative example help to illustrate the mechanism, and we also described their characteristics. For the hydrogel actuator, which is just an example that we found conveniently and that meets the above characteristics. We clarify that any rigid or soft actuator with the above characters can be applied to the actuation of high-performance water surface jumping robots based on the proposed principle.

We have edited the manuscript on page 13, line 239 shown as follows,

“The hydrogel actuator exhibits a much faster energy release time of 0.6 ms, compared to 2.1 ms for hard springs and 1 ms for soft springs due to the fracture driven power amplification mechanism as we presented in previous work⁴¹, details of the further optimization of the hydrogel actuator can be found in Supplementary Note 4.”

We have further added the above-mentioned optimizations and corresponding discussion in Supplementary Information as Supplementary Note 4.

2. To assess the hydrogel actuator's suitability for repeated use in microrobotics, it would be helpful if the authors could clarify how swelling impacts its performance, specifically in terms of response time, force output, and energy efficiency. Additionally, has the authors conducted tests on performance stability across multiple jumping cycles to ensure durability and consistent functionality? Providing this information would strengthen understanding of the actuator's reliability for continuous applications.

The authors appreciate the reviewer's comment. As the reviewer's suggestion, we tested the swelling stability of the hydrogel actuator in water (Supplementary Fig. 3, as shown in Figure R2.4). Multiple hydrogel actuators were placed in water at the same time, and the volume changes were observed after taking them out at regular intervals. As shown in Figure R2.4, the hydrogel actuator experienced limited volume expansion within 6 hours due to the water absorption. In addition, we also tested the weight change of the hydrogel actuator as shown in Supplementary Fig. 4 (Figure R2.5), which shows the hydrogel actuator only experienced limited water absorption and swelling in the water environment. The possible reason is that the P(VP-co-AA) hydrogel has a high degree of cross-linking that causing the polymer chains with high density (Supplementary Fig. 2b and 2c), which limits the entry of water molecules and reduces the ability of water absorption. In addition, the high proportion of VP in the monomer may also reduce the overall hydrophilicity. Due to its limited water absorption capacity, i.e., there is no significant change in volume and weight within 6 hours. We believe that within the usage scenario of contact with water for a short period, its swelling ability has limited influence on the performance of the hydrogel actuator.

Figure R2.4 (Supplementary Fig. 3). The top view and side view image of the actuators after water absorption over different time periods.

Figure R2.5 (Supplementary Fig. 4). The weight change of the actuator after water absorption over different time periods.

We have added the swelling test data as Supplementary Fig. 3 and Fig. 4 and corresponding discussion in Supplementary Note 2 shown as follows:

“In addition, P(VP-co-AA) hydrogel shows the dense network structure endows its mechanical performance as shown in Supplementary Fig. 2b and 2c. We have tested the water absorption ability in a water environment (Supplementary Fig. 3 and Fig. 4), the limited volume and weight change within 6 h indicates the excellent anti-swelling properties, which potentially originate from the high-density polymer chains induced by a high degree of cross-linking that limits the entry of water molecules.”

We clarify that the hydrogel actuator is single-use because it outputs energy in a destructive manner. The phase change is irreversible and the energy release is realized by fracture at the bottom surface. When repeated actuation is performed, it is difficult to store a large amount of energy similar to the first-time actuation due to the lack of bottom constraints. We tested its repetitive actuation by repairing the actuator. Specifically, the hydrogel precursor was injected into the phase transition region after first actuation by syringe and then polymerized by UV light. Although we achieved the secondary water surface jumping motion, its performance was much lower compared with the first time jumping as shown in Supplementary Fig. 22 (Figure R2.6). The possible reason is that the internal polymer chain undergoes high-temperature plasticization, and only limited constraints are formed after repair, resulting in reduced energy storage. Although hydrogel actuators are single

time use, we can make up for this shortcoming by preparing them on a large scale. The hydrogel actuator can be prepared on a large scale by simple UV-polymerization method, the actuation strategy is similar to that of bullets in a gun which can be easily replaced. In addition, we clarify that the novelty of the current work lies in the proposed water surface jumping mechanism and design principle for high-performance water surface jumping robot, the actuator demonstrated in the manuscript is a representative example for the actuation of water surface jumping soft robot as it satisfies the design principle of the actuation module. Any other rigid or soft actuator with fast response and high power output can be used based on the proposed strategy.

Figure R2.6 (Supplementary Fig. 22). The water surface jumping height comparison of the first and second actuation of the soft robot.

We have edited the manuscript on page 13, line 250 as follows,

“Consequently, the hydrogel actuator was chosen to demonstrate the new water surface jumping mechanism due to its combination of ultrafast energy release time and high driving force output. The current hydrogel actuator has the shortcoming of single time use despite this can be compensated by simple large-scale fabrication (Supplementary Fig. 22). Furthermore, it is important to note that this actuator serves as a convenient and typical candidate, other rigid or elastic actuation structures with faster response times and greater force-to-mass ratios can also be utilized in jumpers based on the proposed water surface jumping mechanism.”

3. To enhance the robot's versatility and expand its usability for sustained microrobotic applications on or near water surfaces, would the authors consider integrating the jumping mechanism

with additional locomotion modes, such as directional movement or swimming? While the manuscript demonstrates an impressive water surface jumping capability comparable to semi-aquatic arthropods, combining jumping with other movement types could support more complex or continuous operations in water-based environments.

The authors appreciate the reviewer's comment and completely agree with the suggestion that integrating with additional locomotion modes can further expand the application potentials in water-based environments. As the reviewer's suggestion, we have added the demonstration of the water surface soft robot combined with the magnetic navigation and the results were added in Supplementary Fig. 27 (as shown in Figure R2.7) and Supplementary Video 5. Due to human marine activities, oily pollutants in the water surface environment are gradually increasing. The formation of oil film can hinder the reoxygenation of water bodies, affect the growth of plankton, and destroy the ecological balance. As shown in Figure R2.7a and R2.7b, through module structure design, we introduced the magnetic sponge module with magnetic navigation and automatic oil absorption abilities for the construction of water surface oil removal miniature robot. The magnetic sponge module is fabricated from PDMS sponge doped with NdFeB micro particles. Specifically, the soft robot is placed on the water surface and approaches the oily pollutants (doped with red oil-soluble dye for clear display) floating on the water surface under wireless magnetic navigation. The magnetic sponge can automatically and quickly absorb the oil droplets into its body. After all the oil droplets are absorbed, the robot reaches the designated position under magnetic navigation and is activated by NIR light stimulation to jump out of the water surface, thus realizing the oil removal (Supplementary Video 5). As shown in Supplementary Fig. 28 (Figure R2.8), it can be clearly seen that the oily substance is absorbed into the magnetic sponge, thus verifying its potential applications in environmental detection and protection.

Figure R2.7 (Supplementary Fig. 27). (a) Schematic illustration of the water surface jumping soft robot combined with a magnetic navigation module to realize the water surface oily pollutants removal in a constrained volume. (b) The image of the water surface jumping soft robot integrated with a magnetic oily pollutants absorption module. (c-f) A series of images illustrating the oily pollutants absorption process navigated by magnetic field and followed oil removal.

Figure R2.8 (Supplementary Fig. 28). Images of the soft robot and magnetic sponge before and after the oily pollutant absorption.

We have edited the manuscript and added discussion on page 20, line 351 using the above-mentioned figures as follows:

“In addition, we also explored the application potentials in the field of the oily pollutant removal in water surface environment, as shown in Supplementary Fig. 27a-27b and Supplementary Video 5, a PDMS magnetic sponge module was integrated into the soft robot, enabling the soft robot to be navigated by the magnetic field and automatically absorb the floating oily pollutants (Supplementary Fig. 27c-27e). After completing the oily pollutant collection, the soft robot jumps out of the water surface (Supplementary Fig. 27f). The comparison of the PDMS sponge shown in Supplementary Fig. 28 indicates the application potentials in environment detection and protection. These experiments provide an indirect remote-control strategy and validate the application potential of the current water surface jumpers for water environment related applications.”

We further added the fabrication method of the magnetic PDMS sponge in the Method section on page 28, line 526 shown as follows,

“The oil absorption module (PDMS sponge) was fabricated by emulsion method based on the literature⁴⁵. Specifically, 12 g paraffin oil and 10 g NdFeB micro particles were added to the 10 g mixed PDMS solution (10:1) under vigorous stirring. Then, 70 g distilled water was slowly added dropwise under stirring until the PDMS-water emulsion was formed. Next, the PDMS-water emulsion was polymerized in the oven (70 °C for 2 h). After polymerization, the PDMS sponge was washed with ethanol three times and dried in the oven. The magnetic oil absorption module was successfully fabricated after cutting and integrated on the soft robot.

Reviewer #3:

The paper presents an insect-sized water-jumping robot that can jump on a water surface with a high drag. The research topic is interesting, and the results show an impressive jumping performance of the robot. A novel hydrogel-type actuator and hydrophobic structure on the body seem to be the main components for jumping on water. As a potential application, the authors show that the robot takes a micro underwater robot out of the water for recycling. The demonstration of the potential application is also interesting and meaningful.

We thank the reviewer for the careful review and positive feedback on the research topic and demonstration of the potential application. We have carefully considered the concerns raised by the reviewer about the novelty and the significant contributions not being clear enough. We have substantially revised our manuscript by clarifying the contribution, revising the Figures, adding more experimental data to support the statement, and re-editing the paper. Please check the following point-by-point response for details.

However, the significant contribution and novelty of the paper are unclear for the readers to be able to learn and understand the mechanism happening on the water surface.

We are grateful for the reviewer's comments. Here, we try to further elucidate the novelty and uniqueness of our work below. Currently, the reported miniature water surface jumping robots mainly rely on the surface tension to provide actuation force for water-surface jumping motion. However, this mechanism faces an inherent physical constraint: the propulsion force must remain below the threshold required to break the water surface (144 mN/m), the power output of the actuator is strictly controlled in order to avoid the momentum acquisition reduction induced by water surface broken.⁴ In other words, the powerful actuators with considerable force output have been developed but their force output ability is constrained during application by this mechanism. Our design strategy allows the actuator's force output ability to be released without considering the water surface breaking due to the actuation force originating from the solid interaction rather than the reaction force from the water surface. As the reviewer's comment, we agree that the similar design of the hydrogel actuator and superhydrophobic launching pad have been published in the literature. However, for the current design principles that the momentum transmission system (i.e., a split type design using soft tendon connections) is more important and is first demonstrated in the construction of the water surface jumping robots. Previously reported water surface jumping robots

are typically fabricated with rigid materials and an integrated structural design. This design principle tightly combines the power output and energy conduction with the dynamic water interactions. The limitation brought by the surface tension of water must be considered to improve the power of the actuator for better jumping performance.

For the current design principle demonstrated in the manuscript, we pioneered the use of soft materials to build water surface jumping robots. Despite the simple structural design (i.e., soft tendons connection), the initial momentum acquisition and the interaction between the robot and the water surface are separated. The actuation force mainly comes from the solid interaction (the actuator hits the launching pad to generate reaction force) rather than the reaction force from the water surface. This actuation force acquisition strategy allows increasing the power output of the actuator without considering the limitation of the water surface breaking, which maximizes the initial velocity while minimizing the kinetic energy loss due to the water surface broken.

In particular, we clarify that the proposed strategy inspired by biological models is a universal mechanism that can be applied in different water surface robot designs. We outline three important design principles for achieving high-performance water surface jumping in the manuscript, i.e., Superhydrophobic contact with water surface, a momentum transmission system achieved by split structure design, and light weight, powerful actuation module with high F/m ratio (Supplementary Fig. 15, as shown in Figure R3.1). The soft robot demonstrated in the manuscript is only a typical example that satisfies the above-mentioned requirements. Any further designs consistent with the proposed strategy can be utilized to realize high-performance water surface jumping motion.

Furthermore, for the current design of the water surface jumping soft robot shown in the manuscript, we still consider it represents a clear advance in the field of water surface jumping. To the best of our knowledge, there is only one water surface jumping robots fabricated with soft materials which is actuated by magnetic field.⁹ However, the robots motion is constrained inside the volume of the coil that used for the magnetic field production. The reason why there are so few water surface jumping soft robots is that the soft materials, unlike rigid materials (i.e., springs and motors), has inherent limitations that prevent them from storing large amounts of energy in the network structure and have a slow response time.^{10, 11} These characteristics conflict with the requirements of traditional water surface jumping motion. Based on the proposed strategy, we use a hydrogel actuator with power amplification effect and soft tendon connection to achieve excellent performance that is comparable to natural water surface jumping. Although some modules have been studied, we still believe that the current design is an advancement in the field of water surface jumping robot.

Figure R3.1 (Supplementary Fig. 15). The key design principles of the proposed water surface jumping mechanism.

We have added the above discussion as Supplementary Note 6 for clearer description and revised the manuscript on Page 20, line 364 shown as follows:

“In summary, we have introduced a water surface jumping strategy beyond the surface tension dominated mechanism for insect-scale robots that harnesses three key design principles to break through the physical constraints of current jumping mechanism (Supplementary Note 6), including a superhydrophobic body for floating on water surface, a light-weight powerful actuation module that can provide adequate propulsion force in an ultrashort time (muscle activation timescale) for large initial momentum acquisition, and a soft momentum transmission system for highly efficient kinetic energy transfer.”

We further revised Fig. 1 to highlight our proposed strategy by comparison shown as follows,

In terms of the scientific analysis of this research, the paper should describe the design principles that can manage the drag from the water surface such as shape, force profile, and actuation speed, for achieving the desired performance for the robot. The maximum performance of the robot is recorded, but the dynamics of the mechanism are not fully established so that we can understand the design objective. As far as I understand from this paper, it seems that the high power of the actuator is the main design principle.

The authors appreciate the reviewer's comment. First of all, we want to clarify that the high power of the actuator is only one of the design principles. The construction of high-efficient momentum transmission system is also important. The soft connection structural design makes the acquisition of actuation force mainly rely on the interaction of solids (i.e., rebound force when the actuator hits the launching pad) rather than the reaction force of the surface tension of water (i.e., $< 144 \text{ mN/m}$) utilized in traditional surface tension-dominated jumping mechanism, thereby allowing the power output of the actuator to be increased as much as possible without considering the water surface

broken. Secondly, the design principles of the high-performance water surface jumping robot we proposed are briefly described in the following according to their importance classification:

(1) High-efficient momentum transmission system. A split-type design using soft tendon connections ensures efficient momentum transfer. This approach facilitates the momentum matching between the high-velocity actuation module (enabled by the substantial bounce force) and the low-velocity launching pad (due to the water resistance), maximizing energy transfer efficiency.

(2) Lightweight and powerful actuators. The soft connection allows a significant increase in the actuator's power output, making the actuator capable of providing adequate propulsion force (F/m ratio) in an ultrashort time that allows the jumper to acquire momentum for high take-off velocities.

(3) Superhydrophobic contact with water. Note that the superhydrophobic body shows less importance in the high-performance water surface jumping motion. It can make the soft robot float stably on the water surface in the initial state. Other methods that achieve this can be used to build water surface jumping robot.

We have added a discussion on the design principle for our proposed water surface jumping mechanism on page 10, line 168 shown as follows,

“In surface tension-dominated jumping, the reaction force from the water surface (i.e., driving force) must remain below the surface tension threshold to maximize the kinetic energy transfer to the jumper instead of the water by avoiding the water surface broken, which can result in significant splashing. Once the water surface is broken, the drag force from the disrupted water leads to substantial kinetic energy loss for the jumper. In contrast, the new mechanism we propose does not require the water surface to remain unbroken during the jumping process. The soft connection structural design makes the acquisition of actuation force mainly rely on the interaction of solids (i.e., rebound force when the actuator hits the launching pad) rather than the reaction force of the surface tension. Thus allows the significant increase of the actuator's power output for initial momentum acquisition. These design principles allow greater flexibility in energy transfer, enhancing overall performance without the constraints imposed by traditional designs.”

We agree that the dynamics of the mechanism should be provided for helping the reader's understanding of the design principles. To address the reviewer's suggestion, we have tested the influencing factors of the jumping performance as shown in Figure R3.2 (also shown as Supplementary Fig. 14). The corresponding detailed discussion have been added in Supplementary information as Supplementary Note 5.

Figure R3.2 (Supplementary Fig. 14). The influence factors of the jumping performance. (a) The jumping height and response time of water surface jumping motion for soft robot irradiated with different light intensity. (b) The jumping height of the water surface jumping motion for soft robot with different graphene content. (c) Schematic illustration of the graphene content inducing the difference of phase transition volumes inside the actuator's body. (d) The jumping height of water surface jumping motion for soft robot and stress-strain curve of hydrogel with different (d, e) water content and (f, g) MBA content, respectively. (h) The jumping height and contact area with different diameters of pad. (i) The jumping height of the soft robot with different (i) softness and (j) length of the tendon. Note that the data are captured with the light intensity is 2.8 W ($4.4 W/cm^2$).

For the current design of the soft robot, the jumping performance is influenced by the energy input (light intensity), energy production (graphene content), force output of the hydrogel actuator (mechanical properties of the hydrogel actuator) and the properties of the launching pad and soft tendon.

As shown in Supplementary Fig. 14a (Figure R3.2a), the increased light intensity results in the faster thermal production and deeper penetration, which induce a larger phase transition volume inside the hydrogel actuator and more vapor accumulation before fracture, thus increasing the force output of the hydrogel actuator, i.e., the more energy input, the more energy conversion and output. The composition of the hydrogel actuator influences the force output of the actuation module. Specifically, the transformed thermal to trigger the phase transition originates from the photothermal effect of the graphene. The jumping performance increased first as the graphene content increases from 0.07 wt% to 0.17 wt%, the more and faster thermal production and higher intensity steam explosion enabled by the increased graphene content induce better jumping performance (Figure R3.2b). However, the excess graphene content (i.e., the graphene content more than 0.17 wt%) leads to the decreased penetration depth of NIR light due to the blocking effect, which induces the smaller phase transition volume (Figure R3.2c) and lower force output exhibited as the reduction of the jumping height. The concentration of the graphene content needs to be carefully selected in order to balance the trade-off between the photothermal energy conversion efficiency and phase transition volume.

The mechanical property of the hydrogel also affects the force output through changing the energy accumulation ability before fracture. The more strain energy accumulation through mechanical property design during deformation process (i.e., phase transition inside the actuator before fracture), the larger force output at the moment of fracture and the better actuation performance. Specifically, the water content and crosslinker (MBA) content are changed to tailor the properties of the hydrogel actuator. Figure R3.2d and the representative stress-strain curve in Figure R3.2e shows that the water content through influencing the toughness to change the energy storage before hydrogel fracture, the difference in energy storage ability results in the exhibited tendency of jumping performance (i.e., first increase and then decrease), the best jumping performance is observed in the water content of 60 wt% means that the biggest energy accumulation before fracture. The effect of the crosslinker content on the jumping performance exhibits the similar trend, as presented in Figure R3.2f and R3.2g, the best jumping performance can be realized of the hydrogel actuator conclude the MBA content is 0.23 wt%.

In addition to the influence factors from the force output of the hydrogel actuator, the superhydrophobic structure of the PDMS pad and the properties of tendon can also affect the jumping performance. We have measured the influence of the PDMS pad's diameter on the jumping performance of the soft robot, a larger contact area induces greater surface tension means that the stronger resistance (the diameter from 7 mm to 13 mm), in other words, the soft robot can suffer bigger force output from the actuator before sinking that leads to the bigger velocity difference between the

PDMS pad and actuation modular to obtained a higher take-off velocity. However, the excess contact area (i.e., > 13 mm) increases the gravity of the soft robot and the resistance (i.e., surface tension as resistance when lifting) during the jumping process that induced the rapid degradation of the jumping performance (Figure R3.2h). Moreover, the soft tendon is indispensable for high performance water surface jumping motion, the jumping robot without soft tendon or through rigid material connections cannot realize the high-efficient momentum transfer inside the robot's body, resulting in unsuccessful jumping or low jumping performance. The mechanical properties of the soft PDMS tendon (i.e., the length and softness) also have influences on the water surface jumping performance. We performed the vertical water surface jumping motion of the soft robot with tendons have different stiffness, which were fabricated by changing the ratio of the monomers and the hardeners of PDMS (as shown in Figure R3.2i). The stiffness of the tendon should be chosen carefully as it provides different reaction force ratios for impact.¹² For example, a too-soft tendon fabricated with Ecoflex will increase energy dissipation due to the large deformation during the stretching process, the suitable tendon stiffness is fabricated of PDMS-10 that can realize the best jumping performance. Furthermore, the jumping performance is also influenced by the length of the soft tendon shown in Figure R3.2j, the jumping performance shows limited difference for the water surface jumping soft robot with the tendon length less than 3 cm, and it decreases rapidly as the length continually increase. Potential reason is that the actuation module of the soft robot moves upward due to the reaction force generated by the fractured hydrogel actuator hitting the PDMS pad, which results in the initial acceleration caused by the instant impulse and subsequent deceleration motion. The length of the tendon should match with the displacement distance of the acceleration stage to obtain the best jumping performance.

We further revised the manuscript on page 14, line 275 shown as follows,

“For the current design, the jumping performance of the soft robot can be influenced by many factors, e.g., light intensity, power output of the hydrogel actuator, hinge length and diameter of the launching pad, etc. We have tested these influencing factors and the detailed discussion can be found in Supplementary Note 5”

In terms of technical improvement, the actuation mechanism with hydrogel is used as the main actuator, which is introduced in the previous study. The hydrophobic structure on the body was also published in multiple papers. A breakthrough enabling technology for this study is unclear. I understand many engineering techniques should be applied to make this robot, but it is not enough to convince of the significance of the research to be published in Nat. Comm. Before going

to the detailed review of the manuscript, these major concerns should be convinced to consider the publication of the paper. The research topic and approach are good, but scientific findings and technical improvement are required for the broad readership of the journal.

The authors appreciate the reviewer's comment and positive feedback for the research topic and approach. We would like to emphasize that our study is fundamental research of the water surface jumping mechanism inspired by nature rather than report a miniature robot. The soft robot demonstrated in the manuscript is a representative example that helps to illustrate the design principles for high-performance water surface jumping robot. The reported miniature water surface jumping robot to data are mainly relying on the surface tension dominated mechanism, i.e., the actuation force originates from the reaction force from the surface tension of water. However, this mechanism faces an inherent physical constraint: the propulsion force must remain below the threshold required to break the water surface (144 mN/m), the power output of the actuator is strictly controlled in order to avoid the momentum acquisition reduction induced by water surface broken. The way to improve jumping performance is to control the force output infinitely close while less to the surface tension of water or reduce the self-weight of the robot. The water surface jumping mechanism we proposed provides unique way to improve the robot's jumping performance, i.e., unlimitedly increase the power output of the actuator within the materials tolerable range to obtain the greatest possible initial momentum, which realized by the high-efficient momentum transmission system that separate the process of the actuator's force output and the dynamic interaction with water. It means that we do not need to consider the momentum loss caused by the water surface breaking because the actuation force mainly originates from the solid interaction rather than surface tension. We believe that the proposed water surface jumping mechanism can inspire more designs and development of the water surface jumping robot.

In addition, for the representative design of the soft robot that we demonstrated in the manuscript, as the reviewer's comment, we agree that the similar design of the hydrogel actuator and superhydrophobic launching pad have published in the literature. As mentioned above, for the current design principles that the momentum transmission system (i.e., a split type design using soft tendon connections) is more important and first demonstrated in the construction of the water surface jumping robots. In addition, to the best of our knowledge, except for the magnetic driven PDMS robot reported by Wang et. al,⁹ all miniature water surface jumping robots are made of rigid materials, which utilize the springs and other components to store energy. Currently, there is no soft robot that can achieve high-performance water surface jumping motion. Although our robot design utilizes some reported structural designs, the robot's body fabricated entirely of soft materials based

on the water surface jumping mechanism we proposed and the excellent jumping performance indicating it still represent an advancement in the current field of soft robots, which will promote new environmental interactions and potential application areas for soft robots. We believe that design principles and water surface jumping mechanism we proposed can serves as a reference for the development of various water surface soft robots.

We have added the above discussion in Supplementary Information as Supplementary Note 6 and revised the Fig. 1 in manuscript to highlight the novelty and contributions of this work.

References

1. Y. Chen, J. Yang, X. Zhang, Y. Feng, H. Zeng, L. Wang & W. Feng. Light-driven bimorph soft actuators: design, fabrication, and properties. *Mater Horiz* **8**, 728-757 (2021).
2. D.D. Han, Y.L. Zhang, J.N. Ma, Y.Q. Liu, B. Han & H.B. Sun. Light-mediated manufacture and manipulation of actuators. *Adv. Mater.* **28**, 8328-8343 (2016).
3. W.G. Yang, X.W. Wang, Z. Wang, W.F. Liang & Z.X. Ge. Light-powered microrobots: Recent progress and future challenges. *Opt. Laser. Eng.* **161**, 107380 (2023).
4. J.S. Koh, E. Yang, G.P. Jung, S.P. Jung, J.H. Son, S.I. Lee, P.G. Jablonski, R.J. Wood, H.Y. Kim & K.J. Cho. Jumping on water: Surface tension-dominated jumping of water striders and robotic insects. *Science* **349**, 517-521 (2015).
5. J.S. Koh, S.P. Jung, R.J. Wood & K.J. Cho. A jumping robotic insect based on a torque reversal catapult mechanism. *2013 IEEE/RSJ International Conference on Intelligent Robots and Systems*, 3796-3801 (2013).
6. F. Jiang, J. Zhao, A.K. Kota, N. Xi, M.W. Mutka & L. Xiao. A miniature water surface jumping robot. *IEEE Robot. Autom. Lett.* **2**, 1272-1279 (2017).
7. Y.F. Chen, H.Q. Wang, E.F. Helbling, N.T. Jafferis, R. Zufferey, A. Ong, K. Ma, N. Gravish, P. Chirarattananon, M. Kovac & R.J. Wood. A biologically inspired, flapping-wing, hybrid aerial-aquatic microrobot. *Sci. Robot.* **2** (2017).
8. M. Ilton, M.S. Bhamla, X.T. Ma, S.M. Cox, L.L. Fitchett, Y. Kim, J.S. Koh, D. Krishnamurthy, C.Y. Kuo, F.Z. Temel, A.J. Crosby, M. Prakash, G.P. Sutton, R.J. Wood, E. Azizi, S. Bergbreiter & S.N. Patek. The principles of cascading power limits in small, fast biological and engineered systems. *Science* **360**, eaao1082 (2018).
9. Y. Wang, X. Du, H. Zhang, Q. Zou, J. Law & J. Yu. Amphibious miniature soft jumping robot with on-demand in-flight maneuver. *Adv. Sci.* **10**, e2207493 (2023).
10. M.T. Li, X. Wang, B. Dong & M. Sitti. In-air fast response and high speed jumping and rolling of a light-driven hydrogel actuator. *Nat. Commun.* **11**, 3988 (2020).
11. B. Hao, X. Wang, Y. Dong, M. Sun, C. Xin, H. Yang, Y. Cao, J. Zhu, X. Liu, C. Zhang, L. Su, B. Li & L. Zhang. Focused ultrasound enables selective actuation and Newton-level force output of untethered soft robots. *Nat. Commun.* **15**, 5197 (2024).
12. P.M. Wensing, A. Wang, S. Seok, D. Otten, J. Lang & S. Kim. Proprioceptive actuator design in the MIT cheetah: Impact mitigation and high-bandwidth physical interaction for dynamic legged robots. *IEEE Trans. Robot.* **33**, 509-522 (2017).

Reviewer #1:

The jumping mechanism is now quite clearly explained and supported by Supplementary Figs. 6 -7. Please explicitly indicate the cases where F_{\max}/m and not F/m ratio was presented.

I like all the new figures. They do help to understand the volume of experimental work performed by the authors.

Comment 15: Second part of Eq. 8 is wrong (see attached equation). It seemed that you've used diameter instead of radius in the equation for the truncated cone volume. The actual volume is 4 times smaller. Eqs. 10 and 12 give the upper bound of the potential energy of displaced water.

I have no further comments.

We thank the reviewer for the positive comment and careful reviewing of our revised manuscript. We have addressed the comments and concerns in this round and made corresponding changes in the revised version. Please review the details of the point-by-point responses below.

1. Please explicitly indicate the cases where F_{\max}/m and not F/m ratio was presented.

The authors appreciate the reviewer's comments. As the reviewer's suggestion, we have edited the Table 1 and Fig. 3d in the manuscript, Supplementary Note 1 and Note 4 in the Supplementary Information, with specific expressions of \$F_{\max}/m\$, details can be found in revised manuscript and supplementary information.

2. Comment 15: Second part of Eq. 8 is wrong (see attached equation). It seemed that you've used diameter instead of radius in the equation for the truncated cone volume. The actual volume is 4 times smaller. Eqs. 10 and 12 give the upper bound of the potential energy of displaced water.

We thank the reviewer for careful review and for pointing this out. We have revised the calculation process and recalculated the energy loss and the jumping energy efficiency, the details shown as follows:

“Similarly, as shown in Supplementary Fig. 9, the volume of the splashing water column can be roughly estimated by considering it as a hollow truncated cone, the thickness of the truncated cone can be estimated by the transparency distribution in high-speed camera screenshots, the weight of the splashed water is ~ 3.63 g can be calculated as follows,

$$m_2 = \rho(V_{outer} - V_{inner}) \quad (11)$$

Where ρ is the density of water, V_{outer} and V_{inner} are the volume of the outer and inner truncated cone, respectively, which can be calculated by $V = \frac{1}{3}\pi h(R^2 + Rr + r^2)$ based on the parameters given in Supplementary Fig. 9.

Therefore, the potential energy of the splashed water above the water surface is 0.79 mJ that can be roughly estimated by,

$$E_2 = m_2gh_2 \quad (12)$$

The energy loss during the dynamic interaction with water is calculated to be 2.49 mJ.”

In addition, the jumping efficiency was revised shown as follows:

“Therefore, the released energy applied to the PDMS pad of the hydrogel actuator for the jumping actuation can be expressed as:

$$E_{explosion} = E_s + E_b + E_{jumping} \quad (15)$$

Based on this, the released energy calculated through equation (15) is 5.16 mJ. And the energy conversion efficiency can be calculated through equation (16) is 49.22 %.

$$\eta = \frac{E_{jumping}}{E_{explosion}} \quad (16)''$$

More details can be found in Supplementary Note 3.

Reviewer #2:

Authors addressed and discussed the raised issues in the light of the reviewer's comments. In the revised manuscript, the authors emphasized explanation of the insect-scale water surface jumping mechanisms for soft robots. In addition, they have adequately supplemented with references and requested data to support their claims. The manuscript now demonstrates a more comprehensive view of optimizations in terms of structure design, swelling stability of the actuator, and demonstration of the robot providing detailed information. I recommend the paper could be published in Nature Communications without further change.

We sincerely thank the reviewer for the positive comments.

Reviewer #3:

Authors addressed issues raised by the reviewer.

Here are some comments that can improve the quality of the paper.

- **In Fig.1, the illustration of the springtail jumping sequence is redundant information in this research. It is hard to find relevance between this mechanism and the springtail jumping.**
- **The authors say that the soft connection is a key design factor for the robot. It should be proven with theoretical modeling for a clear understanding how the stiffness of the connection affects to the jumping performance.**
- **Is it possible to give any design optimization for the high jumping on the water surface based on the power of the actuator?**

We thank the reviewer for the careful reviewing of our revised manuscript and providing following comments and concerns. We have addressed these comments and concerns and made corresponding changes in the revised version. Please review the details of the point-by-point responses below.

1. In Fig.1, the illustration of the springtail jumping sequence is redundant information in this research. It is hard to find relevance between this mechanism and the springtail jumping.

We are grateful for the reviewer's comments. We have carefully considered the reviewer's concerns and try to further elucidate the connections of the water surface jumping kinetics between springtails and our miniature soft robot. The biomechanical model of the springtail's water-surface jumping motion has been explained in detail.^{1, 2} We designed high-performance water surface jumping robot by referring to the bottom principles of springtails and with further engineered optimizations. The detailed explanations are shown as follows:

(1) Surface tension utilization (i.e., floating in the initial state). The springtails possess a light body mass and super hydrophobic epidermis, allowing them to establish superhydrophobic contact with the water surface and stable floating without sinking. Based on this principle, we have fabricated the PDMS pad with superhydrophobic micro structure for the water surface jumping soft robot, which utilized to achieve stable floating in the initial state.

(2) Actuation. Springtails have a special catapult organ, i.e., the furcula, which actuated by rapid muscle contraction to hit the water surface to acquire driving force, thus overcoming the surface tension of the water to detach with water surface. The ultrafast muscle activations (approximately

< 1 ms for springtails as shown in Fig. 1b in the manuscript) allows the efficient initial momentum acquisition. Based on this principle, the basic selection standard of the actuator for the water surface jumping robot is the ultrafast response speed (e.g., 0.6 ms of the selected hydrogel actuator in this work). In addition, springtails exhibit the highest F/m ratio (2.92×10^3 N/kg) among semi-aquatic arthropods capable water surface jumping motion. The current hydrogel actuator was selected due to its remarkable F/m value (i.e., 1.34×10^4 N/kg), which is two orders of magnitude higher than that of previously reported engineered water surface jumping robots.

(3) Energetics and kinematics. Through evolution, the organisms have developed optimal kinematic characteristics to adapt to their environments. The water surface jumping motion of springtails is actuated by the catapult organ, with subsequent adjustments by the body system to achieve high momentum transfer efficiency. Based on this object to improve the jumping performance, we designed a split-type design utilizing soft tendon connections which ensures efficient momentum transfer.

Although the appearance of our designed robot differs significantly from that of springtails, we have incorporated the springtail-inspired bionic designs into working principles and underlying mechanisms as illustrated in Fig. 1. We emphasize that our study focusses on fundamental research of the water surface jumping mechanism inspired by nature rather than report a miniature robot. The soft robot demonstrated in the manuscript serves as a representative example that helps to illustrate the design principles for high-performance water surface jumping robot. We believe that the proposed design principles and water surface jumping mechanism can advance the development of miniature soft robotics for water-environment-related applications.

2. The authors say that the soft connection is a key design factor for the robot. It should be proven with theoretical modeling for a clear understanding how the stiffness of the connection affects to the jumping performance.

The authors appreciate the reviewer's comment. The "soft connection" described in this article is defined in contrast to the "rigid connection" that utilized in almost all previously reported water jumping robots.³⁻⁹ The rigid connections are typically fabricated from metal or other hard materials to form joints that integrate various components of the robot into a system. This design requires the robots to rely entirely on surface tension to provide driving force during the water surface jumping motion. However, this surface tension-dominated mechanism faces an inherent physical constraint:

the propulsion force must remain below the threshold required to break the water surface, which limiting the actuator power output and the efficient momentum acquisition.

In contrast, we pioneered the use of soft material connections to break through the physical constraint of water surface dominated mechanism. In our design, the actuation force mainly comes from the solid-solid interaction (i.e., the actuator hits the launching pad to generate a reaction force) rather than relying on the reaction force from the water surface. This actuation force acquisition strategy allows increasing the power output of the actuator without considering the limitation of the water surface breaking, which maximizes the initial velocity while minimizing the kinetic energy loss due to the water surface broken. We have experimentally demonstrated the difference in jumping performance between rigid connection and soft connections, the results were shown in Fig. 1f (as shown in Figure R3.1) and Supplementary Fig. 19 (as shown in Figure R3.2). This represents the core design principle and novelty of our work.

Figure R3.1 (Fig. 1f). The comparison of the robot with soft connection and rigid connection.

Figure R3.2 (Supplementary Fig. 19). (a) Photograph of the robot use direct connection method and (b) series of images illustrating the corresponding unsuccessful water surface jumping motion. (c) Photograph

of the robot with rigid connection method and (d) series of images illustrating the corresponding low performance water surface jumping motion.

In addition, although theoretical modeling of the influence of stiffness on jumping performance extends beyond the scope of the innovations presented in our paper, we have, as the reviewer's suggestion, added a simplified theoretical model to address the reviewer's concern regarding the effect of hinge stiffness on jumping performance. As shown in Figure R3.1, the jumping robot during actuation can be simplified as a spring-oscillator model that the masses are connected by springs of varying stiffness.

Figure R3.3. The schematic illustration of the simplified spring-oscillator model.

The dynamic model of the robot's body (m_b) can be expressed as follows:

$$m_b \frac{d^2 x_b}{dt^2} + 2k(x_b - x_p) + C \left(\frac{dx_b}{dt} \right)^2 = F(t) \quad (1)$$

Where $F(t)$ is the impulse response function, x_b and x_p is the displacement of the robot's body and launching pad, respectively. C is the damping coefficient can be calculated as follows,

$$C = \frac{C_d \rho \pi r_0^2}{2} \quad (2)$$

Where C_d is the drag coefficient during the flying motion in the air. ρ is the density of air (1.20 kg/m³). r_0 is the radio of the robot's body.

The stiffness k in the equation (1) can be expressed as,

$$k = \frac{E \cdot \omega \cdot d}{l} \quad (3)$$

Where E is the elastic modules of the hinge, and the ω , d , l is the thickness, width and length of the hinge, respectively.

The initial motion condition is

$$x_b |_{t=0} = 0 \quad (4)$$

$$\frac{dx_b}{dt} |_{t=0} = v_0 \quad (5)$$

Where v_0 is the initial speed of the robot's body.

The stiffness influences of the soft hinge can be theoretical estimated with the initial speed of the robot's body based on the Equation (1). In addition, we also further tested representative soft connections with different stiffnesses as shown in Supplementary Fig. 14i, which were consistent well with the theoretical model.

3. Is it possible to give any design optimization for the high jumping on the water surface based on the power of the actuator?

The authors appreciate the reviewer's comment. We speculate that the reviewer's comment is whether the power output of the actuator can be regulated to influence the robot's water jumping performance. **The answer is affirmative. The power output of the hydrogel actuator can be roughly controlled by adjusting the amount of energy input and modifying the mechanical properties of the hydrogel network. As the reviewer's suggestions, we tested these influencing factors and identified the parameters that maximize power output. In addition, we also provide suggestions on possible optimization directions, including actuation method and actuator's directional motion control.**

(1) Parameter optimization for improved jumping performance. As shown in Supplementary Fig. 14a (Figure R3.4a), the increased light intensity results in the faster thermal production and deeper penetration, which induce a larger phase transition volume inside the hydrogel actuator and more vapor accumulation before fracture, thus increasing the force output of the hydrogel actuator, i.e., the more energy input, the more energy conversion and output. The composition of the hydrogel actuator influences the power output. Specifically, the transformed thermal to trigger the phase transition originates from the photothermal effect of the graphene. The jumping performance increased first as the graphene content increases from 0.07 wt% to 0.17 wt%, the more and faster thermal production and higher intensity steam explosion enabled by the increased graphene content induce better jumping performance (Figure R3.4b). However, the excess graphene content (i.e., the graphene content more than 0.17 wt%) leads to the decreased penetration depth of NIR light due to the blocking effect, which induces the smaller phase transition volume (Figure R3.4c) and lower force output exhibited as the reduction of the jumping height. The concentration of the graphene content needs to be carefully selected in order to balance the trade-off between the photothermal energy conversion efficiency and phase transition volume.

The mechanical property of the hydrogel also affects the force output through changing the energy accumulation ability before fracture. The more strain energy accumulation through mechanical

property design during deformation process (i.e., phase transition inside the actuator before fracture), the larger force output at the moment of fracture and the better actuation performance. Specifically, the water content and crosslinker (MBA) content are changed to tailor the properties of the hydrogel actuator. Figure R3.4d and the representative stress-strain curve in Figure R3.4e shows that the water content through influencing the toughness to change the energy storage before hydrogel fracture, the difference in energy storage ability results in the exhibited tendency of jumping performance (i.e., first increase and then decrease), the best jumping performance is observed in the water content of 60 wt% means that the biggest energy accumulation before fracture. The effect of the crosslinker content on the jumping performance exhibits the similar trend, as presented in Figure R3.4f and R3.4g, the best jumping performance can be realized of the hydrogel actuator conclude the MBA content is 0.23 wt%.

Therefore, the power output of the hydrogel actuator can be adjusted by applying suitable light energy input and changing the materials composition. The highest jumping performance can be realized of the hydrogel actuator conclude graphene content, water content and MBA content is 0.17 wt%, 60 wt% and 0.23 wt%, respectively, which actuated with the light intensity is 10 W/cm².

Figure R3.4 (Supplementary Fig. 14). The influence factors of the jumping performance. (a) The jumping height and response time of water surface jumping motion for soft robot irradiated with different light intensity. (b) The jumping height of the water surface jumping motion for soft robot with different graphene content. (c) Schematic illustration of the graphene content inducing the difference of phase transition volumes inside the actuator's body. (d) The jumping height of water surface jumping motion for soft robot and stress-strain curve of hydrogel with different (d, e) water content and (f, g) MBA content, respectively. Note that the data are captured with the light intensity is 2.8 W (4.4 W/cm²).

(2) Suggestions for jumping direction control optimization. In the current work, our robot achieves directional jumping motion by changing the hinge length ratio to introduce asymmetry. A potential optimization strategy for directional control is to create asymmetry in the actuation force by controlling the phase-change position of the hydrogel actuator. As shown in Figure R3.5, the actuator is fabricated with transparent hydrogel as the main body. By embedding graphene-based actuation modules at specific positions relative to the center of mass, asymmetric and directional actuation can be achieved. This simple structural design optimization strategy allows for the preparation of hydrogel actuators in various shapes, tailored to the robot's morphology rather than being limited to cylindrical forms. Such flexibility could enable additional functionalities and multimodal control of the robot.

Figure R3.5. Schematic illustration of the jumping direction control optimization.

(3) Suggestions for trigger method. In this work, the power output of the hydrogel actuators is triggered through high-intensity laser induce the phase transition inside the hydrogel. Although the light driven actuation is an efficient, clean and pollution-free strategy, it has inherent disadvantages, such as the light is easily blocked and the limited penetration depth. For potential application scenarios such as confined spaces, tethered actuation methods may offer better adaptability and more advantages compared to light driven approaches. Based on the actuation principle of the hydrogel actuator (i.e., phase transition induces rapid deformation to accumulate strain energy and instant release), any method capable of triggering the phase change within the hydrogel actuator can be utilized to achieve the actuation of soft robot. For example, embedding a resistance wire at the desired phase-change location within the actuator's body allows temperature increase through electric field excitation, triggering the phase change and subsequent deformation and power output. Exploring alternative trigger methods may significantly expand the application fields of the current soft jumping robot.

We have added the parameter optimization strategy of the hydrogel actuator in Supplementary Information. The corresponding detailed discussion can be found in Supplementary Note 5.

References

1. V. Ortega-Jimenez, E. Challita, B. Kim, H. Ko, M. Gwon, J.S. Koh & M.S. Bhamla. Directional takeoff, aerial righting, and adhesion landing of semiaquatic springtails. *Proc. Natl. Acad. Sci. U.S.A.* **119**, e2211283119 (2022).
2. A.A. Smith & J.S. Harrison. Jumping performance and behavior of the globular springtail *dicyrtomina minuta*. *Integr Organism Biol* **6**, obae029 (2024).
3. J.S. Koh, E. Yang, G.P. Jung, S.P. Jung, J.H. Son, S.I. Lee, P.G. Jablonski, R.J. Wood, H.Y. Kim & K.J. Cho. Jumping on water: Surface tension-dominated jumping of water striders and robotic insects. *Science* **349**, 517-521 (2015).
4. M. Gwon, D. Kim, B. Kim, S. Han, D. Kang & J.S. Koh. Scale dependence in hydrodynamic regime for jumping on water. *Nat. Commun.* **14**, 1473 (2023).
5. D.L. Hu, M. Prakash, B. Chan & J.W.M. Bush. Water-walking devices. *Exp Fluids* **43**, 769-778 (2007).
6. J. Zhao, X. Zhang, N. Chen & Q. Pan. Why superhydrophobicity is crucial for a water-jumping microrobot? Experimental and theoretical investigations. *ACS Appl. Mater. Interfaces* **4**, 3706-3711 (2012).
7. B. Shin, H. Kim & C. K. Towards a biologically inspire small-scale water jumping robot. *2008 2nd IEEE RAS & EMBS International Conference on Biomedical Robotics and Biomechatronics*, 127-131 (2008).
8. Y.F. Chen, H.Q. Wang, E.F. Helbling, N.T. Jafferis, R. Zufferey, A. Ong, K. Ma, N. Gravish, P. Chirarattananon, M. Kovac & R.J. Wood. A biologically inspired, flapping-wing, hybrid aerial-aquatic microrobot. *Sci. Robot.* **2** (2017).
9. J.H. Yan, K. Yang, T. Wang & J. Zhao. Research on design and jumping performance of a new water-jumping robot imitating water striders. *2015 IEEE International Conference on Information and Automation*, 353-358 (2015).

The manuscript entitled „Beyond surface tension-dominated water surface jumping” presents an explosive steam/water blast driven jumping of a polymer block from a platform floating on the water surface and connected elastically to the block. The polymer lifter flies up with the platform from the water surface to the height of up to 18 body lengths after it has been heated during several seconds with a near infrared light source. Authors demonstrated the crucial role of superhydrophobicity of the launching disc platform for the unit takeoff and significant influence of the softness of connection between the platform and the lifter on the lifting height, stated that the lifter unit outperform a cylindrical metal coil spring in propulsion force to mass ratio. The possibility of the flight control in horizontal direction was demonstrated. It was achieved by shortening of one of the two elastic band connecting the lifter and the platform. Finally, authors have shown a possible application of the proposed device for removal of microrobots from water. The manuscript is written short and clear.

My general considerations:

It is not fair to compare mechanically driven jumping insects with engineering systems using jet propulsion. It is similar to comparison of an airplane with a bird. Even a 7 m jump of dolphins *Tursiops truncatus* could be outperformed by flight of a small rocket launched even directly from the water surface.

To the disadvantages of the proposed system I would list the necessity of an external energy source to evaporate some water in the polymer block and its' single-shot usage. It shouldn't be also an imperative to remove the launching platform from water.

The lifting height of the unit is indeed primary depends on drag and buoyant forces and only partially depends on surface tension, but its' stationary floating on the water surface still depends on surface tension forces, since the platform should be superhydrophobic for the successful unit flight.

According to Fig. 3d, the propulsion force of the proposed unit should be $5 \cdot 10^4 \cdot 0.4 \cdot 10^{-3} = 20$ N. This *considerably* differs from the highest value presented in the manuscript (e.g. Fig. 1, Tab. 1). Besides, even if acceleration, $F/m = 1.34 \cdot 10^4$ (Tab. 1), was acting during 0.6 ms (Line 97), then the take-off velocity should be 8.25 m/s. That was obviously not the case.

Not self-consistent results regarding jumping speed: Fig. 1e, Fig. 1d(III), Fig. 2, Fig. 3b, Fig. 4e, Supplementary Tab. 1; and height: Fig. 2, Fig. 4d, Fig. 4f, Supplementary Tab. 1, Supplementary Fig. 8a.

Further comments:

No light source power is given.

Page 3, Line 57: “0.5 ms”, the muscle contraction is never so fast. Just read the reference you provided carefully.

Page 4, Line 81: “... the long actuation duration ($\sim 10^1$ milliseconds)...”, however, actuation time for Motor-spring is 2 ms, according to Supplementary Tab. 1.

Page 4, Lines 82-83: "... fast muscle strokes ($\sim 10^0$ millisecond)..." , however, actuation time for e.g. fisher spider is 52 ms, according to Supplementary Tab. 1.

Page 5, Lines 99-100: "... due to the incompressibility of water (Fig. 1e).", water is incompressible in both situations presented in top and bottom images. Buoyant force, conventional and wave drag forces propose the required for the lift up reaction force.

Page 5, Lines 104-105: "... rigid connections 104 would result in momentum mismatch and hinder the initial velocity acquisition...", the actual mechanism is that the stem/water stream coming out of the lifter hits the platform. And momentum of the stream is transferred to the platform, which is almost equal and opposite to that of the lifter.

Fig. 2b: Water strider should be instead of "Water spider".

Fig. 3.: Figures should appear after they first mentioned. **c** and **d**, it should be noticed, if the results are obtained by launching the actuators from hard substrate and not from the water surface.

Eq. 1: Both m_b and m_p or their ratio should be given.

Page 11, Lines 185-186: "This increased velocity difference between v_b and v_p results in a more pronounced upward movement of whole jumper..." , the more the velocities difference, the less the total velocity, according to Eq. 1.

Page 11, Lines 199-201: "The released energy by the hydrogel actuator applied to the floating pad and the jumping energy utilization efficiency are estimated to be 3.97 mJ and 63.97 %, respectively (see calculation details in Supplementary Note 3).", based on the size of the lifter in Supplementary Fig. 6 and Eq. 7, E_b should be ~ 100 times larger.

Page 13, Line 243: "... from the water surface at a constant speed of 0.1 mm/s.", the work of adhesion significantly depends on the pull-off speed, which is in m/s range during "jumping experiments". Besides, the work of adhesion measured by authors is ~ 30 times smaller than the estimated by them explosion energy.

Page 25, Lines 451: Any black dye could be used in the jumper instead of graphene, I suppose.

Page 27, Lines 493: Please, specify the actual power and not the power density of the laser diode illuminating the lifter.